# Targeting TRIP13 in favorable histology Wilms tumor with nuclear export inhibitors synergizes with doxorubicin
Karuna Mittal[1,2,14], Garrett W. Cooper[1,2,14], Benjamin P. Lee [1,2,14], Yongdong Su [1,2], Katie T. Skinner[1,2], Jenny Shim [1,2,3], Hunter C. Jonus[1,2], Won Jun Kim[4,5], Mihir Doshi[4,5], Diego Almanza [4,5], Bryan D. Kynnap[4,5], Amanda L. Christie[4], Xiaoping Yang [5], Glenn S. Cowley [5], Brittaney A. Leeper [6], Christopher L. Morton[7], Bhakti Dwivedi[3], Taylor Lawrence[1,2], Manali Rupji [3], Paula Keskula[5], Stephanie Meyer[8], Catherine M. Clinton[8], Manoj Bhasin [1,2,3], Brian D. Crompton [5,8], Yuen-Yi Tseng[5], Jesse S. Boehm[5], Keith L. Ligon [5,9,10], David E. Root [5], Andrew J. Murphy [7], David M. Weinstock [4,5,8,13], Prafulla C. Gokhale [6], Jennifer M. Spangle[3,11], Miguel N. Rivera [5,12], Elizabeth A. Mullen [8], Kimberly Stegmaier [5,8], Kelly C. Goldsmith [1,2,3], William C. Hahn [4,5] ✉ & Andrew L. Hong [1,2,3] ✉

Wilms tumor (WT) is the most common renal malignancy of childhood. Despite improvements in the overall survival, relapse occurs in ~15% of patients with favorable histology WT (FHWT). Half of these patients will succumb to their disease. Identifying novel targeted therapies remains challenging in part due to the lack of faithful preclinical in vitro models. Here we establish twelve patient-derived WT cell lines and demonstrate that these models faithfully recapitulate WT biology using genomic and transcriptomic techniques. We then perform loss-of-function screens to identify the nuclear export gene, *XPO1*, as a vulnerability. We find that the FDA approved XPO1 inhibitor, KPT-330, suppresses *TRIP13* expression, which is required for survival. We further identify synergy between KPT-330 and doxorubicin, a chemotherapy used in high-risk FHWT. Taken together, we identify XPO1 inhibition with KPT-330 as a potential therapeutic option to treat FHWTs and in combination with doxorubicin, leads to durable remissions in vivo.

Wilms tumor (WT) is the most common childhood renal tumor and represents ~6% of all pediatric cancers with a peak age of presentation at 3 years[1–3]. In the United States, African American children have 2.5 times higher rates of WT when compared to Caucasian American or Asian American children[4,5]. Approximately 95% of WT patients present with favorable histology (FHWT), whereas the other 5% present with diffuse anaplastic histology (DAWT) which is usually associated with *TP53* mutations or deletions. Current therapy for low risk FHWT patients (Stage I

and II) includes the use of surgery with or without chemotherapy (e.g., vincristine, dactinomycin). For high-risk FHWT disease (e.g., Stage III and IV individuals with pulmonary metastasis or invasion of the renal sinus and capsule), doxorubicin and radiation therapy are added to low-risk disease therapy. Despite increases in response and survival over the past 50 years, ~15% of patients with WT recur[6–8] and salvage regimens which include doxorubicin for low risk patients are successful only in 50% of patients and carry significant morbidity.

[1]Department of Pediatrics, Emory University School of Medicine, Atlanta, GA, USA. [2]Aflac Cancer and Blood Disorders Center, Children's Healthcare of Atlanta, Atlanta, GA, USA. [3]Winship Cancer Institute, Emory University, Atlanta, GA, USA. [4]Department of Medical Oncology, Dana-Farber Cancer Institute, Boston, MA, USA. [5]Broad Institute of MIT and Harvard, Cambridge, MA, USA. [6]Experimental Therapeutics Core and Belfer Center for Applied Cancer Science, Dana-Farber Cancer Institute, Boston, MA, USA. [7]Department of Surgery, St. Jude Children's Research Hospital, Memphis, TN, USA. [8]Department of Pediatric Oncology, Dana-Farber Cancer Institute, Boston, MA, USA. [9]Department of Pathology, Brigham and Women's Hospital, Boston, MA, USA. [10]Department of Oncologic Pathology, Dana-Farber Cancer Institute, Boston, MA, USA. [11]Department of Radiation Oncology, Emory University School of Medicine, Atlanta, GA, USA. [12]Department of Pathology, Massachusetts General Hospital, Boston, MA, USA. [13]Present address: Merck & Co., Rahway, NJ, USA. [14]These authors contributed equally: Karuna Mittal, Garrett W. Cooper, Benjamin P. Lee. ✉e-mail: william_hahn@dfci.harvard.edu; andrew.hong2@emory.edu

A major limiting factor in testing novel targeted therapies in WT is the lack of faithful in vitro preclinical models. Prior cell line models of WT have been recharacterized as other pediatric cancers such as rhabdoid tumor (e.g., G401) and Ewing sarcoma (e.g., SK-NEP)[9–12]. Recent efforts to generate WT cell lines have been limited due to finite passaging[13–16]. Despite this limitation, a repository of WT organoids and patient-derived xenografts (PDXs) has been developed in recent years[17–19].

Nomination of rational therapeutics for organoid or in vivo PDX studies, however, requires systematic in vitro efforts using faithful cancer cell lines. Here, we have developed faithful cell line models using genome and transcriptome sequencing of WT which recapitulate known WT biology. We then performed functional genomic screens focused on druggable targets to nominate WT therapeutics[20–22].

## Results
### WT cell lines faithfully recapitulate genomic and transcriptomic features of WT

Recent studies have shown the feasibility of generating short-term WT cell lines with limited genomic testing[13–16]. We developed 12 short-term WT cell lines from 10 patients (10 patient-derived cell lines and 2 PDX-derived cell lines following one passage of the PDX in mice (Annotated as T2); Fig. 1; "Methods"). Ten patient samples were obtained at time of diagnosis and two were obtained at time of recurrence (Aflac_2377T and CCLF_PEDS_0002T hereafter PEDS_0002T). Nine patients had favorable histology Wilms tumor (FHWT) and one patient had diffuse anaplastic Wilms Tumor (DAWT; CCLF_PEDS_0023T hereafter PEDS_0023T).

We then performed ultra-low coverage whole genome sequencing (WGS) to infer copy number status, whole exome sequencing (WES) to identify known mutations in WT, and RNA-sequencing to assess the transcriptome in the patients' tumor and the matched cell lines. We identified 1q gain, a poor prognostic factor in Wilms tumor biology, in two patients (Aflac_2597 and PEDS_0023) and we identified combined loss-of-heterozygosity (LOH) in 1p and 16q in two patients (Aflac_2315 and Aflac_2365)[23,24]. We further observed gain in chromosome 12, which has been associated with relapse, in two patients (Aflac_2597 and

CCLF_PEDS_0041 hereafter PEDS_0041). Methylation status was not assessed in this study. We then assessed the mutational profiles of these tumors and cell lines and observed genetic similarity between patient tumor samples and matched cell lines (Fig. 1, Supplementary Tables 1 and 2 and Supplementary Data 1). We found that the observed mutations reflected the spectrum of mutations seen in the WT samples profiled in the National Cancer Institute (NCI) Therapeutically Applicable Research to Generate Effective Treatments (TARGET)[25] as well as other studies[18,26] (Fig. 1). We identified genetic heterogeneity between the tumor and the tumor derived cell line in 5 of the 10 patient samples (Aflac_2315, Aflac_2377, Aflac_2597, PEDS_0023, and PEDS_0041), supporting the previously observed genetic heterogeneity in Wilms tumor samples[18,27]. Lastly, we identified 2 of the FHWT patient-derived tumor cell lines had no mutations typically observed in Wilms tumor, an observation also seen in 5% of patient samples in prior genomic analyses of FHWT[25].

To further confirm that our cell lines were consistent with Wilms tumor, we performed RNA-sequencing of these samples and compared it to the TARGET[25] and St. Jude Children's Research Hospital's WT datasets[18] ("Methods") using uniform manifold approximation and projection (UMAP)[28]. We observed that most of our WT and normal cell lines clustered closely with the WT tumor and normal samples, respectively (Fig. 2a). However, one FHWT cell line (Aflac_2315) did not clearly cluster with our tumor tissue and cell lines. SIX2 is elevated in Wilms Tumor and the mean log2 counts for our normal tissue and cell lines were 2.54 (standard deviation of 0.32) whereas in our FHWT was 3.73 (standard deviation of 0.20). Aflac_2315 had SIX2 log2 counts of 3.86 in the tumor and 3.39 in the cell line. More broadly, we observed that SIX2 and CITED1 were generally upregulated across our WT cell lines, further consistent with WT biology (Fig. 2b)[29–31]. Interestingly, a gene which has a known therapeutic target, XPO1, was modestly upregulated across renal tumors (Fig. 2b). Although the tumor cell lines exhibited similar expression of XPO1 as compared to tumor tissue, we also found that the normal cell lines included in this study also had upregulation of XPO1. Collectively, our findings suggest that our patient-derived cell lines recapitulate known biology of WT and serve as faithful representations of WT.

**Fig. 1 | Overview of the genomic analysis of Wilms tumor samples.** Co-mutation plot representing the clinicopathological information (top panel), loss of heterozygosity and copy number analyses (middle two panels), and mutations (bottom panel) in the matched WT cell lines, PDX-derived cell lines, patient tumors, and normal cell lines and tissues where applicable. Each column represents a particular sample.

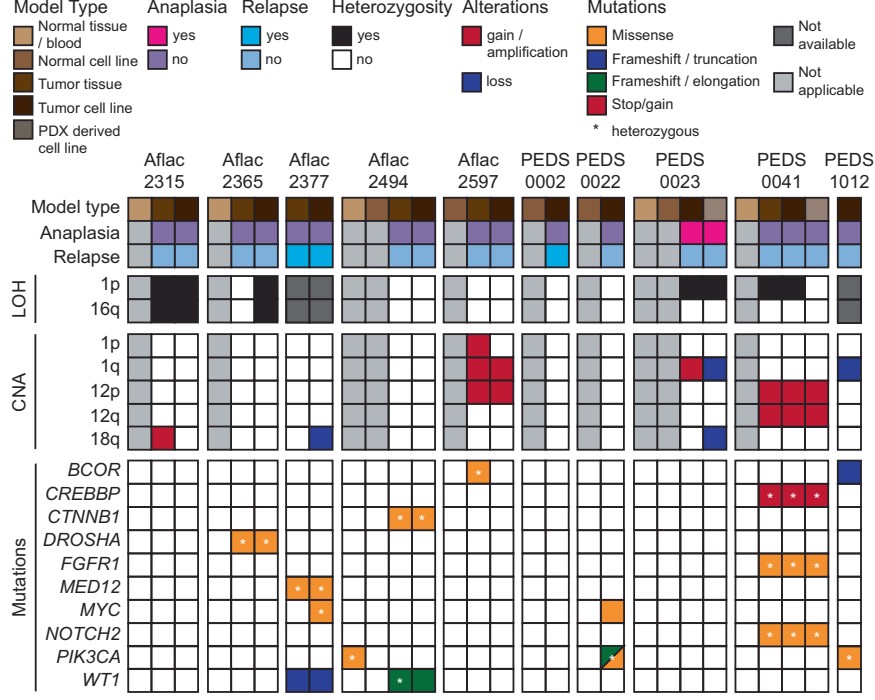

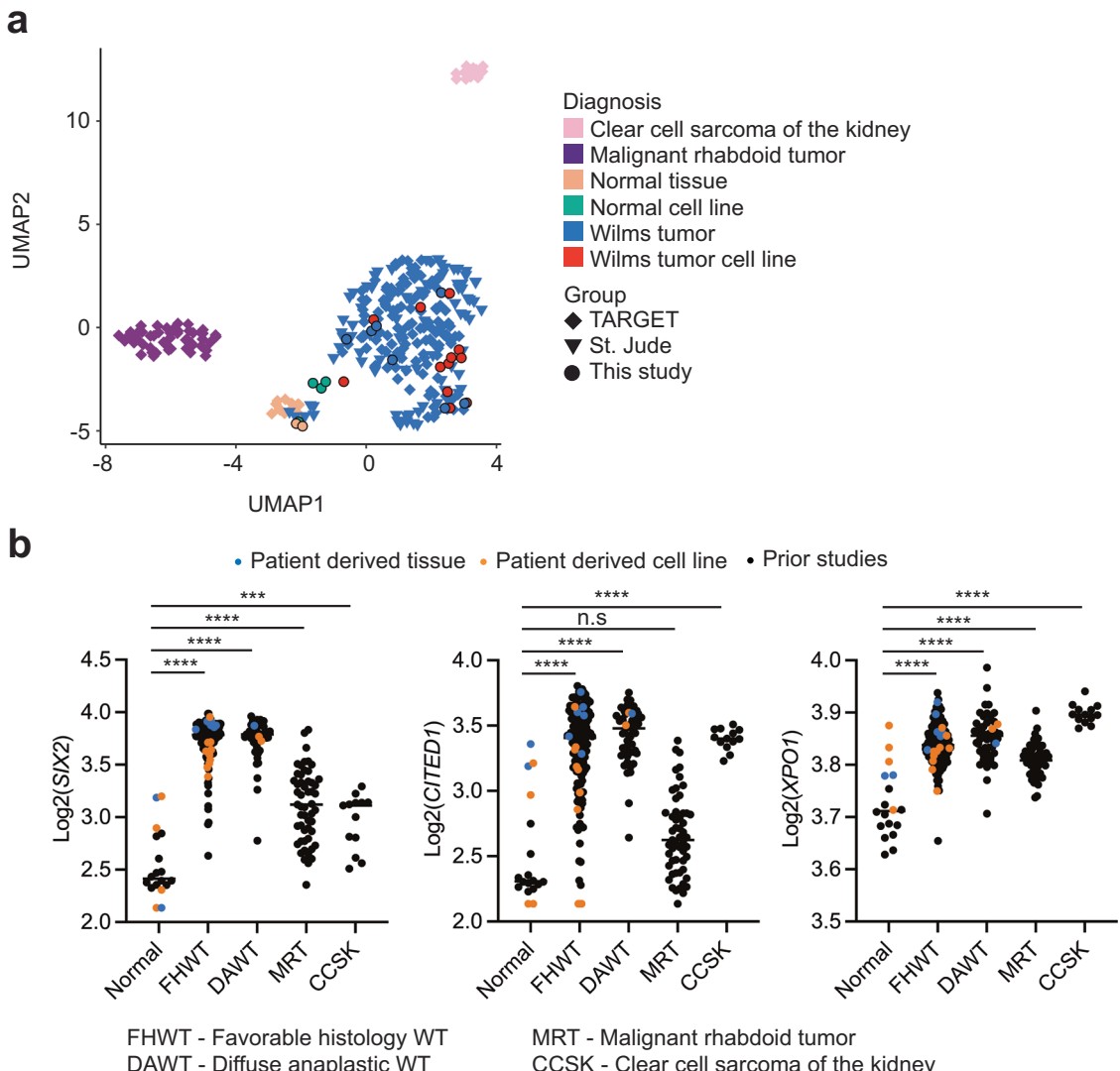

**Fig. 2 | Overview of the transcriptomic analysis of Wilms tumor samples. a** Two-dimensional representation of RNA-seq data using uniform manifold approximation and projection (UMAP) demonstrates high concordance between TARGET (diamond, $n = 205$) and St. Jude primary WT (triangles, $n = 80$) and samples in this study (circles, $n = 25$). Normal tissue samples, clear cell sarcoma of the kidney, and malignant rhabdoid tumor all clustered separately. **b** Dot plots of normalized read counts representing the higher expression of commonly dysregulated genes (*SIX2*, *CITED1*, and *XPO1*) in WT in the TARGET ($n = 205$) and St. Jude ($n = 53$) datasets and this study ($n = 25$) with known sub-diagnosis. Blue and orange dots are samples included in this study. Black bars indicate the mean of each group.

## RNAi and CRISPR-Cas9 screens identify XPO1 as a potential therapeutic target

We then asked if we could identify genetic vulnerabilities in WT despite the short-term lifespan of these cell lines. Following rapid expansion within the first five passages, we subjected cell lines derived from three unique patient samples (PEDS_0002, PEDS_0023, and PEDS_0041) to targeted loss of function RNA interference and CRISPR-Cas9 screens ("Methods" and Supplementary Data 2). Given the limited number of cells that could be expanded, we used the Druggable Cancer Targets (DCT) library consisting of 429 genes that focused on known or upcoming therapeutic targets that had small molecule inhibitors[22,32] (Fig. 3a).

For the RNAi screens, we introduced the DCT lentiviral library into PEDS_0002T, PEDS_0041_T1, and PEDS_0041_T2 cells and then used Model-based Analysis of Genome-wide CRISPR-Cas9 Knockout (MAGeCK) to identify 20 genes which were required for survival across these patient-derived cell lines[33] (Fig. 3b and Supplementary Fig. 1a–d). We subsequently performed CRISPR-Cas9 screens in PEDS_0002T, PEDS_0023_T1 and PEDS_0023_T2 to identify 24 genes that when deleted led to decreased viability across these cell lines (Fig. 3b and Supplementary

Fig. 1e–k). From these orthogonal screens, we identified seven genes which overlapped between RNAi and CRISPR-Cas9 screens. These included genes involved in nuclear export (*KPNB1* and *XPO1*), regulators of the cell cycle (*KIF11* and *POLA1*), DNA damage (*UBA1* and *DDB1*), and cell survival (*BIRC5*) (Fig. 3b).

We focused on the role of the nuclear export inhibitor KPT-330 (selinexor) due to its recent FDA-approval in multiple myeloma and diffuse large B-cell lymphoma[34]. Further, KPT-330 has recently been identified as a potential effective therapy which targets renal tumors with aberrant XPO1 activation[35]. We first assessed the expression levels of *XPO1* across 85 cancer types through the University of California Santa Cruz Treehouse Childhood Cancer Initiative[36]. From 12,719 patient samples, we found 85.4% of Wilms Tumor samples were in the top 20% of samples with high *XPO1* levels (Fig. 3c). To validate the requirement of XPO1 in WT cell lines, we assessed cell viability of WT cells using KPT-330 and *XPO1* suppression with RNAi (Fig. 3d, e and Supplementary Figs. 2a–c and 3). Specifically, we determined the $IC_{50}$ values for our WT cell lines and compared these to normal kidney cell lines. Among tumor cell lines we observed an average $IC_{50}$ of 1.79 μM ± 1.8, with over half of these cell lines

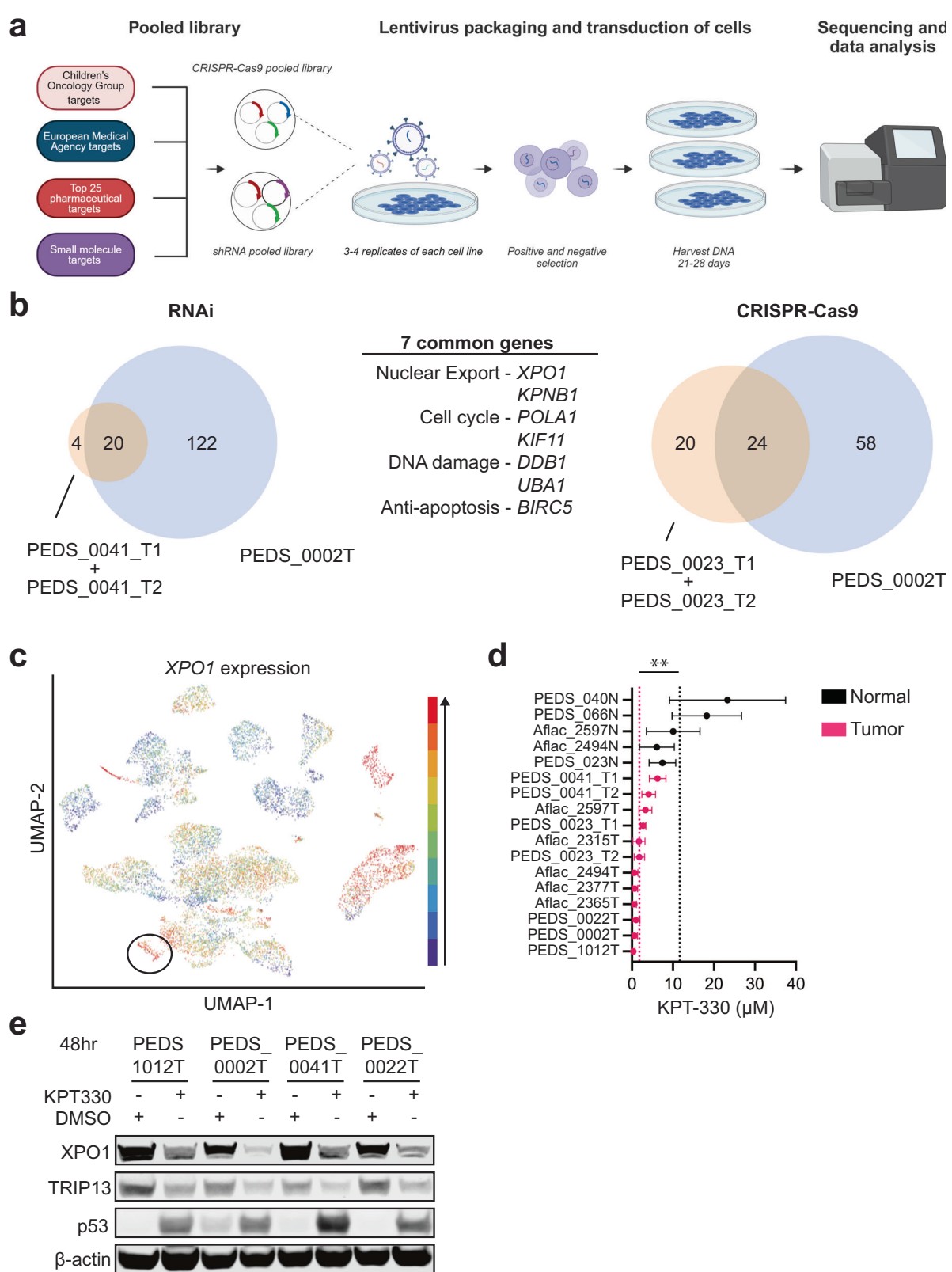

showing an IC$_{50}$ at nanomolar concentrations (e.g., 25–800 nM). Among normal cell lines we observed a 6-fold increase in the average IC$_{50}$ of 11.64 µM ± 6.6 with a *P* value of 0.0002 (Fig. 3d). Moreover, we observed increased sensitivity to KPT-330 in all tumor cell lines for the three matched tumor-normal pairs (PEDS_0023, Aflac_2494 and Aflac_2597). The increased sensitivity of tumor cells as compared to normal cells suggests

KPT-330 may be a selective inhibitor with limited off-target toxicity. We observed that *XPO1* expression was not correlated with KPT-330 sensitivity ($r^2 = 0.004137$, Supplementary Fig. 4a) suggesting that on target activity is not entirely dependent on elevated *XPO1* transcript levels. However, we found that the relative fold change in *XPO1* levels was significantly higher in tumor cell lines as compared to normal cell lines

**Fig. 3 | XPO1 is a potential therapeutic target in Wilms tumor cells. a** Schematic outlining the methodology of CRISPR-Cas9 and RNAi functional screens. Created with BioRender.com. **b** RNAi suppression in three cell lines (patient-derived PEDS_0041_T1 and PDX-derived PEDS_0041_T2 cell lines are grouped together) identified 20 common genes which were critical for the survival of WT cells. CRISPR-Cas9 screens identified 24 common genes in three cell lines (patient-derived PEDS_0023_T1 and PDX-derived PEDS_0023T_T2 cell lines are grouped together), which were critical for the survival of WT cells. Seven genes overlapped between the RNAi and CRISPR-Cas9 loss-of-function screens. These seven genes can be categorized under their role in nuclear export, cell cycle, DNA damage, and apoptosis. **c** UCSC Treehouse transcriptional data from 12,719 samples showing expression of *XPO1* in all tumor types with WT samples circled in black. Blue to red colors signify expression levels with red being the highest among this cohort. **d** Forest plot representing the mean IC$_{50}$ of KPT-330 in the panel of WT cell lines and normal cells (ending with N). SD shown from at least two biological replicates. **\*\*P** value < 0.005 from a Student's two-tailed unpaired *t* test. **e** Immunoblots depicting the decrease in total protein levels of XPO1 and TRIP13 upon treatment with KPT-330. Data shown are representative of two biological replicates.

(*P* value 0.044; Supplementary Fig. 4b). These findings are consistent with prior findings that suggest on target activity of KPT-330 decreases the abundance of XPO1 protein (Fig. 3e and Supplementary Fig. 4c, d), inducing a positive feedback loop which increases XPO1 mRNA levels (Supplementary Fig. 4b)[37–41]. However, the downstream mechanisms of action vary by cancer type. Taken together, these findings suggest that XPO1 is a potential selective therapeutic target in WT.

## XPO1 inhibition induces cell death through the TRIP13/p53 axis

We next investigated a potential mechanism of action for the nuclear export inhibitor KPT-330 in WT. We performed fluorescence-activated cell sorting analyses to assess the changes in the cell cycle following KPT-330 treatment. We found that changes in G1 were not consistent across our FHWT cell lines (Fig. 4a and Supplementary Table 3). However, we saw decreases of S phase and significant increases in G2/M suggesting that KPT-330 in our FHWT cell lines led primarily to a G2/M arrest (Fig. 4a). We focused our efforts on understanding the mechanisms in FHWT. To understand the transcriptional changes driving this G2/M arrest, we treated p53 wild-type PEDS_0041_T1 with DMSO or KPT-330 using the IC50 concentrations (e.g., 6 μM) for 24 hours and performed RNA-sequencing ("Methods"). We then performed differential expression analyses and found 1120 genes differentially expressed (Fig. 4b and Supplementary Data 3). We examined gene sets enriched or suppressed upon KPT-330 treatment using Gene Set Enrichment Analyses (GSEA)[42] and observed 32 hallmark gene sets significantly enriched or suppressed (Supplementary Data 4).

We found gene sets suppressed affecting the progression of G1 (e.g., E2F) and G2/M (Fig. 4c) along with activation of the p53 pathway (Supplementary Data 4). We then confirmed the activation of p53 by observing accumulation of p53 following treatment with KPT-330 by 48 hours (Fig. 3e and Supplementary Figs. 4c–e and 5a). KPT-330 is known to affect tumor suppressor genes and transcription factors such as p53 through nuclear accumulation of p53 and prevention of MDM2/4 related degradation[43]. Since KPT-330 has been associated with upregulating p53 activity, we sought to determine if p53 was indeed important in mediating WT cell death in the context of nuclear export inhibition[44]. We depleted *TP53* using CRISPR-Cas9 in CCLF_PEDS1012T (hereafter PEDS1012T) and CCLF_PEDS_0022T (hereafter PEDS_0022T) as compared to a control gRNA to *LacZ* (Supplementary Fig. 5b). We then assessed the IC$_{50}$ of KPT-330 and found a 12- to 20-fold increase in these values when *TP53* was deleted (Fig. 4d and Supplementary Fig. 5c). This significant change in IC$_{50}$s suggests a critical role of *TP53* in KPT-330 induced cell death in FHWT.

We subsequently assessed the 1120 genes differentially expressed following KPT-330 treatment (Fig. 4b) in the FHWT cell lines. Notably, we found expression of *TRIP13* (Thyroid Hormone Receptor Interactor 13) to be downregulated in KPT-330 treated cells. TRIP13 has been associated with G2/M arrest[45]. In addition, TRIP13 previously was found to interact with co-factors of p53 in injured renal epithelial cells[46] and more recently, has been identified as a cancer predisposition gene in WT[47].

We then focused on evaluating the functional role of TRIP13 in our FHWT cell lines. At the basal level, pediatric renal tumors generally had a modest and significant increase in expression of *TRIP13* (Supplementary Fig. 5d) as compared to adjacent normal kidney controls. We then suppressed the expression of *TRIP13* with RNAi in p53 wild-type FHWT cells using two different *TRIP13* shRNA constructs ("Methods"). To mitigate off-target effects, we used a seed control to one shRNA construct and an RFP non-targeting control[48]. Our cell viability results showed significant cell death in our cell lines transduced with either *TRIP13* shRNA constructs (14–59% cell viability; Fig. 4e and Supplementary Fig. 5e–g). In addition, when we overexpressed TRIP13 in our FHWT cell lines, we found a modest 15–27% increase in cell counts as compared to a luciferase over-expression control (Supplementary Fig. 5h). Previous observations in patient-derived lymphoblasts from a patient with Wilms tumor which harbor a loss of function mutation in *TRIP13* have reported a decrease in proliferation upon TRIP13 overexpression. Other studies looking at cancers such as glioblastoma, colorectal carcinoma, osteosarcoma, non-small cell lung cancer, and hepatocellular carcinoma have shown that increased *TRIP13* expression led to increased proliferation, migration, and invasion[49–53]. These findings suggest TRIP13 has differing roles in different lineages and contexts. Here, in our tumor cells which do not harbor *TRIP13* mutations, we see that suppression of *TRIP13* leads to decreased viability. We found TRIP13 levels increased in our *TP53* deleted cells and were unchanged upon KPT-330 treatment when compared to a non-targeting control (Supplementary Fig. 5i). Collectively, these results suggest that FHWT (e.g., *TP53* wild-type) cells require TRIP13 for survival and when overexpressed, lead to a modest increase in proliferation.

We then assessed the transcriptional changes seen upon *TRIP13* suppression by RNA sequencing. We used the PEDS_0041_T1 and Aflac_2377T cell lines to determine the consequences of suppressing *TRIP13* as compared to our non-targeting shRNA controls ("Methods"). We found 789 differentially expressed genes (Fig. 4f and Supplementary Data 5). We used GSEA to identify gene sets significantly enriched upon suppression of *TRIP13*. We found the gene sets involved in E2F as significantly downregulated which was confirmed with decreased cyclin D1 levels (Fig. 4f, g, Supplementary Fig. 5e–g, and Supplementary Data 6)[54]. We further found, similar to XPO1 inhibition, that suppression of *TRIP13* is anti-correlated with the G2/M checkpoint (Fig. 4g). These findings suggest that suppression of *TRIP13* acts similarly to treatment with KPT-330.

We subsequently asked how suppression of TRIP13 contributed to the phenotypes seen when FHWT cells were treated with KPT-330. We looked at the overlap between differentially expressed genes from our RNA-sequencing experiments (Fig. 4h and Supplementary Data 7). We found 46 genes upregulated and 20 genes downregulated. Interestingly, upregulated genes *MYCT1*, and *SAMD9* have been implicated as tumor suppressors in other cancers such as hematologic malignancies[55–57]. Furthermore, an over-representation analysis identified an immune response (FDR 1.66e-3). When we evaluated the downregulated genes with *TRIP13* suppression, we found genes involved in the cell cycle such as *ASPM* and *CDCA7*. ASPM is essential for mitotic spindle function in neurons[58] and CDCA7 regulates CCNA2, a cyclin with roles in G1 and G2/M, in esophageal squamous cell carcinomas[59]. This was further confirmed when we performed an over-representation test (FDR 1.94e-2).

In sum, our findings show that treatment with KPT-330 in part leads to suppression of TRIP13 which in turn leads to alterations in the cell cycle.

## KPT-330 and doxorubicin are synergistic in vitro and in vivo

Doxorubicin was added to vincristine and dactinomycin in Stage III FHWT patients in the 1980s to account for higher risk disease (e.g., pulmonary metastasis or tumor rupture)[60]. Despite improved overall survival with the

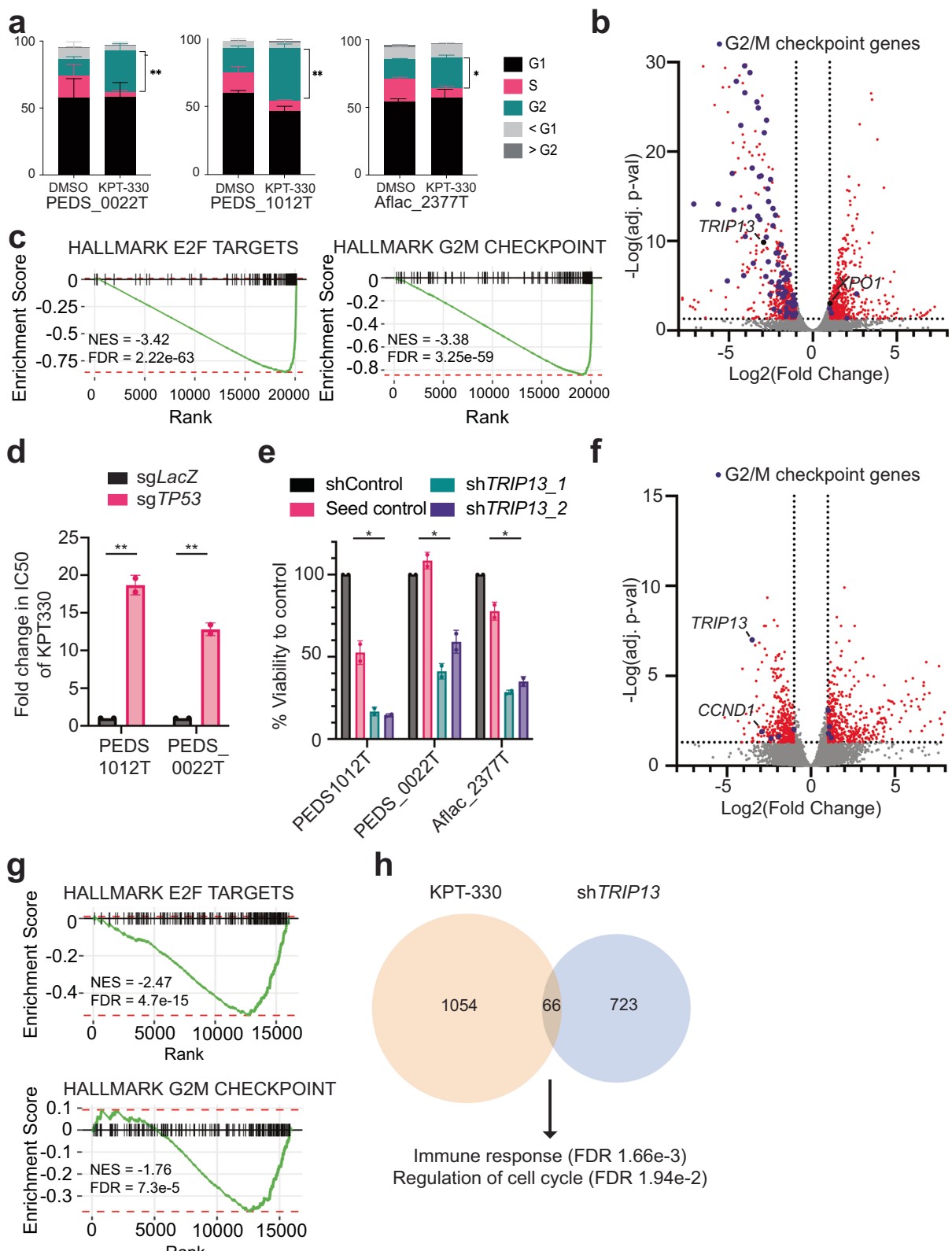

addition of doxorubicin in patients with higher risk FHWT, these cancers recur in 15% of patients[7]. Over the past two decades, only one Phase II clinical trial for relapsed FHWT has been opened (clinicaltrials.gov: NCT04322318). This trial uses a chemotherapy backbone without novel therapeutic targets. Thus, there is a need to identify potential synergistic combination strategies which could be tested in the Phase I/II setting.

We first assessed the role of adding KPT-330 to vincristine, dactinomycin or doxorubicin to determine if there was synergy or additivity with our cell line models of FHWT as measured by CellTiter-Glo. We found the combination of KPT-330 with vincristine or actinomycin with KPT-330 was not synergistic whereas KPT-330 had an additive to synergistic effect with doxorubicin (Fig. 5a).

**Fig. 4 | XPO1 inhibition leads to decreased viability through TRIP13 and p53 axis. a** KPT-330 and DMSO treated PEDS_0022T, PEDS1012T, and Aflac_2377 cells were subjected to cell cycle analysis following flow cytometry. Stacked bar graph representing the proportion of cells across phases of the cell cycle in at least biological replicates and 50k cells counted. Error bars represent mean ± SD. *P value < 0.05, **P value < 0.005 from a Student's two-tailed unpaired t test. **b** Volcano plot representing differential gene expression between the KPT-330 and DMSO treated PEDS_0041_T1 cell line. Scattered points represent genes: the x axis is the fold change for KPT-330 vs. DMSO treated PEDS_0041_T1 cells and the y axis is the P values. Purple dots represent genes in the Hallmark G2/M Checkpoint gene set. **c** Gene set enrichment analysis (GSEA) enrichment score curves for the E2F and G2M hallmark pathways in the PEDS_0041_T1 cells treated with KPT-330. The green curve denotes the NES (normalized enrichment score) curve, the running sum of the weighted enrichment score in GSEA. **d** Deletion of *TP53* significantly increased the IC$_{50}$ of KPT-330 in PEDS1012T and PEDS_0022T. The fold change is based on comparison to a LacZ non-targeting control and based on biological duplicates. Error bars represent mean ± SD, **P value < 0.005 from a Student's unpaired two-sided t test. **e** Change in viability using sh*TRIP13*_1 and sh*TRIP13*_2 across FHWT cell lines as compared to shControl and seed control (to sh*TRIP13*_2). Error bars represent mean ± SD. *P value <0.05 from a Student's unpaired two-sided t test. **f** Volcano plot showing the distribution of significant genes up or down-regulated following sh*TRIP13*. RNA-seq was performed on Aflac_2377T and PEDS_0041_T1 cell lines with sh*TRIP13* as compared to shControl. Biological replicates performed. The x axis is the fold change for sh*TRIP13* vs. shControl cells and the y axis is the adjusted P values. *TRIP13* is downregulated along with *CCND1*. **g** Gene set enrichment analysis (GSEA) enrichment score curves for the E2F and G2M hallmark pathways following suppression with sh*TRIP13*. **h** Number of commonly upregulated and downregulated genes seen in both KPT-330 treated or sh*TRIP13* treated cells. Pathways and significance of overlapping genes obtained from over-representation test.

---

We then evaluated doxorubicin sensitivity in these patient-derived WT cell lines. We found that all WT were sensitive to doxorubicin with IC$_{50}$s in the low nanomolar range (~40–256 nM) as compared to the normal cell lines which had an IC50 range of 131–516 nM (Fig. 5b and Supplementary Fig 6; P value 0.0134). Subsequently, we evaluated the effect of combination KPT-330 and doxorubicin treatment across our cell lines. The combination was synergistic (e.g., Bliss and ZIP scores >10) in 25% of cell lines tested (3 out of 12) across multiple concentrations of KPT-330 and doxorubicin (Fig. 5c). For the remaining 9 tumor cell lines, two of which were DAWT, the combination was found to be additive with synergy scores ranging from 1 to 9. In contrast, this combination was not synergistic in the normal cell lines with scores ranging from −5 to −12 (Fig. 5c). Taken together, we found a synergistic interaction between KPT-330 and doxorubicin in FHWT as compared to normal kidney cell lines.

We followed up these in vitro studies with in vivo studies to further validate the potential synergistic effect of KPT-330 and doxorubicin. Of our patient samples, we were able to generate PDXs from both PEDS_0041 and PEDS_0023. Given that PEDS_0023 has features of anaplastic Wilms tumor, we performed our in vivo studies with the PDX from PEDS_0041, a patient with FHWT. Further the cell line derived from this PDX (PEDS_0041_T2) displayed a synergistic interaction between doxorubicin and KPT-330 (Fig. 5d). Interestingly, the PEDS_0041_T1 cell line and PEDS_0041_T2 were some of the least sensitive WT cell lines to KPT-330 and doxorubicin treatments (Figs. 3d and 5d). We treated tumor xenografts with placebo, doxorubicin, KPT-330, or the combination of doxorubicin and KPT-330 for 28 days and then monitored tumor growth until endpoints were reached or after 150 days following treatment initiation. At 22 days, we compared tumor volume of treatment arms to vehicle. We found that treatment with KPT-330 led to a 49% decrease (P value = 0.046), doxorubicin led to 87% decrease (P value <0.005) and the combination led to 99% decrease (P value < 0.005) in tumor volume with similar findings when comparing log2 fold change (Fig. 5e). We then assessed the effects following cessation of therapy. We found that monotherapy with KPT-330 or doxorubicin led to a median survival of 39 and 53.5 days, respectively (Fig. 5f, g). Further, the combination of KPT-330 and doxorubicin had an undefined median survival with five of eight mice exhibiting a complete response (Fig. 5f, g). Taken together, these observations suggest that inhibition of XPO1 in combination with doxorubicin chemotherapy is synergistic in vivo.

## Discussion

Here, we have developed short-term cell line models of Wilms Tumor that maintain a faithful representation of their genomics and transcriptomics (Figs. 1 and 2). Further, we show that they can be used to perform systematic loss-of-function studies (Fig. 3). We find that integration of targeted RNA interference and CRISPR-Cas9 screens identifies the nuclear export apparatus as a therapeutic target in FHWT.

XPO1 regulates a spectrum of cellular processes by controlling the active transport of over 200 proteins out of the nucleus, including tumor suppressors (TSs) and transcription factors (TFs) (such as TOP2A, p53, WTX, APC, Rb, and PI3K/AKT)[3]. Due to its important physiological role, inhibition of XPO1 by the small molecule inhibitor KPT-330 (selinexor) is widely being tested in phase I and phase II clinical trials in multiple malignancies and has FDA approval in several hematologic malignancies[4,5]. As part of the Pediatric Preclinical Testing Program, KPT-330 was tested as a single agent across a panel of pediatric hematologic and solid tumor PDXs which included several WT xenografts[61]. In this study, the group used KPT-330 in 3 WT PDXs. They found one PDX (KT-10) had maintained a complete response at 42 days post-treatment initiation while the other two had partial responses. However, mechanisms and biomarkers remain unknown.

Here, we show that inhibition of nuclear export leads to primarily a G2/M arrest in FHWT and loss of p53 leads to significant resistance to KPT-330 (Fig. 3). While the role of p53 in response to KPT-330 has been previously reported[62], we identified that suppression or inhibition of nuclear export leads to suppression of TRIP13 in FHWT. We find that although TRIP13 is traditionally thought to affect G2/M[50,63], loss of TRIP13 affects both G1 and G2/M in FHWT (Fig. 4). Furthermore, when we compare the differentially expressed genes between KPT-330 treated cells as compared to cells with *TRIP13* suppression, we find several potential tumor suppressors such as *MYCT1* and *SAMD9*, a gene having antiproliferative properties, whose expression increases. Future studies will explore these findings to better elucidate the function of TRIP13 in FHWT in immune response and cell cycle. Finally, we show that there is significant synergy when KPT-330 is combined with the current standard of care topoisomerase II inhibitor, doxorubicin, in vivo where five of eight mice achieved cure following a single cycle of combination therapy (Fig. 5).

We have observed that the DAWT models used in this study, PEDS_0023_T1 and T2, are sensitive to nuclear export inhibition and have aberrant TP53 function. To discern if nuclear mutant *TP53* accumulation is driving this response in DAWT, we attempted to delete *TP53* using CRISPR-Cas9 similar to PEDS1012T and PEDS_0022T. We observed that these models were unable to tolerate deletion of *TP53*, suggesting that these cells rely on mutant p53 for their proliferation. Together these observations suggest that PEDS_0023_T may have another mechanism through which nuclear export inhibition functions. Further studies with additional models of different p53 mutants are needed to fully characterize the molecular mechanisms driving KPT-330 activity in DAWT.

Finally, there are limitations to our studies. First, these studies utilized short-term patient-derived cell lines. Some features found in patient tumors are likely lost or enriched in the cultured environment, particularly at later passages. Future studies to dissect out the triphasic biology at single cell resolution are needed. In addition, limited expansion of these cell lines prevents high throughput studies across other cell lines included in this study. The use of immortalization techniques may alter our assessment of the biology of WT so identifying growth factors or cytokines to support in vitro growth is needed.

Another limitation of this study is the genetic heterogeneity identified within the tumor and normal samples used in this study. We have transcriptionally confirmed our samples and respective cell lines represent

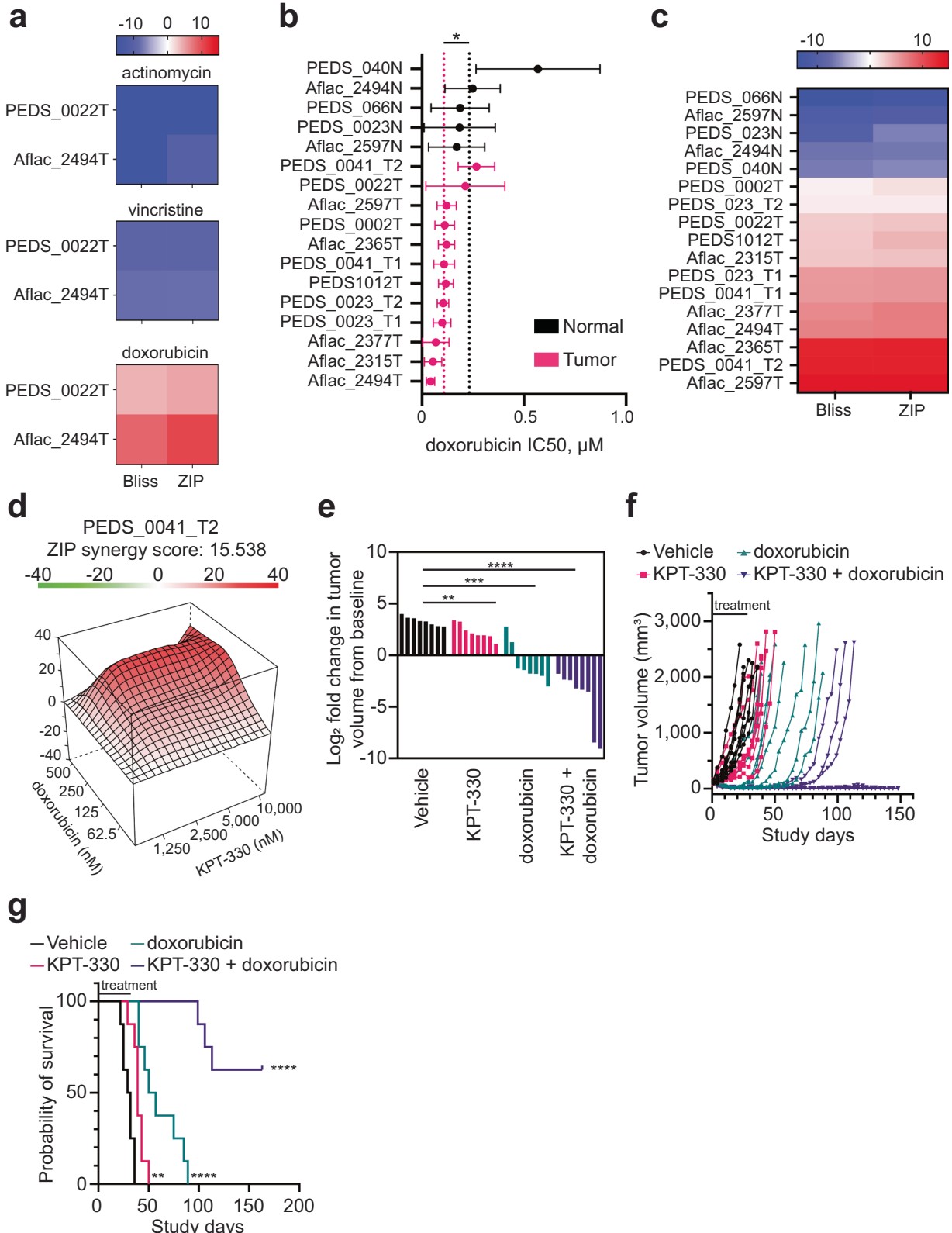

Wilms tumor or normal kidney tissue; however this does not elucidate the mechanisms by which mosaicism and genetic heterogeneity may drive disease pathogenesis in Wilms tumor. Further studies will be needed to assess these features of Wilms tumor in in vitro and in vivo settings.

Taken together, we have identified that KPT-330 acts in part by inhibiting TRIP13 in Favorable Histology Wilms Tumor. In combination with a topoisomerase II inhibitor, nuclear export inhibitors could be studied in patients with high-risk FHWT.

**Fig. 5 | Combination of doxorubicin and KPT-330 is synergistic in vitro and in vivo. a** Heatmap of synergy scores from Bliss and ZIP drug–drug interaction models between KPT-330 with vincristine, actinomycin, or doxorubicin in two FHWT cell lines. Range blue (antagonistic <−10) to red (synergy >10). Representative of at least two biological replicates for each cell line. **b** Mean IC50s for WT cell lines treated with doxorubicin as compared to normal kidney cells. *X* axis is µM. SD shown from at least two biological replicates. *P* value < 0.05 from a Student's two-sided unpaired *t* test. **c** Heatmap showing synergy scores from Bliss and ZIP drug–drug interaction models in WT with KPT-330 and doxorubicin. Range blue (antagonistic <−10) to red (synergy >10). Representative of at least two biological replicates for each cell line. **d** Representative 3D landscape image of the synergy score for CCLF_PEDS_0041_T2 cells treated with doxorubicin (62.5–500 nM) and KPT-330 (1.25–10 µM). **e** CCLF_PEDS_0041 tumor fragments were placed

subcutaneously into the hind flank of NSG (NOD-SCID IL2Rgamma null) female mice. The tumor xenografts were treated with vehicle, doxorubicin, KPT-330 or the combination of doxorubicin and KPT-330 for 28 days. Log fold change of tumor volumes at day 22 were calculated as compared to time of treatment (average 114.3 mm³). **P* value < 0.005, ***P* value < 0.0005, ****P* value <0.00005 from a Student's unpaired two-sided *t* test. **f** Line graph depicting the tumor volume across the study days for each mouse in different treatment groups. Tumor growth was monitored until they met endpoint or the study was terminated at day 150. **g** Kaplan–Meier survival curves representing the probability of overall survival in the patient-derived xenografts (*n* = 8 per treatment group) treated with the vehicle, KPT-330, doxorubicin, and KPT-330 + doxorubicin. **P* value <0.005, ****P* value <0.00005.

## Methods

### Development of patient-derived Wilms tumor cell lines
Tumor cells were isolated from tumor and if available, adjacent normal kidneys tissues of patients with a pathological diagnosis of Wilms Tumor. Samples were obtained with informed consent under IRB approved protocols from patients at either DanaFarber/Boston Children's Cancer and Blood Disorders Center or Aflac Cancer and Blood Disorders Center at the Children's Healthcare of Atlanta and Emory University School of Medicine. Tumor tissue or adjacent normal tissue was finely minced into 1–2 mm³ pieces and cell lines were established using F-media (Supplementary Table 1)[22]. Cells were detached for passaging using TrypLE (Gibco, 12605036). Samples were then plated in six-well plates. Cells were serially passaged after reaching confluency of 70–80%. For PDX cell lines, a tumor fragment (<5 mm in diameter) was implanted subcutaneously into NSG mice and once passaged twice, a tumor fragment was then minced and fragmented on a six-well dish to develop a PDX cell line. In general, Wilms Tumor cells were grown for up to 30 passages and normal kidney cells were grown for approximately 12–15 passages before senescing (Supplementary Table 2). Cell lines were tested for mycoplasma contamination using the MycoAlert Mycoplasma Detection Kit (Lonza, LT07-118) and were negative. All ethical regulations relevant to human research participants were followed.

### Low coverage whole genome sequencing (WGS) and copy number analyses
DNA from tumor tissue, normal tissue if available or blood, and cell pellets from our patient-derived cell lines was extracted (NEB, T3010S). Genomic libraries were prepared (Illumina) by Novogene Inc. and sequenced at 1× on Novoseq 6000 (Illumina). Fastq or BAM files were mapped and aligned using Illumina Dragen v3.7.5 to GrCh38 on Amazon Web Services. Copy number alterations (amplification, gain, deletion) were assessed with ichorCNA[64] and TITAN[65]. Loss of heterozygosity was assessed using TITAN.

### Whole exome sequencing
WES was performed using DNA as described above. For five patients, WES was performed at the Broad Institute Genomics Platform using HiSeq 2000 (Illumina). WES libraries were based on an Illumina Customized Exon (ICE) array. For three patients, WES was performed at Novogene Inc using Novoseq 6000 (Illumina). Libraries were based on an Agilent array. Sample coverage was >100× for tumor and >50× for normal. Fastq or BAM files were mapped and aligned using Illumina Dragen v3.7.5 to GrCh38[66] on Amazon Web Services. Variant call files (vcf) were subsequently processed with OpenCRAVAT v2.4.2 using cancer hotspots, cancer gene landscape, chasm plus, civic, cosmic, mutpanning and NDEX NCI to filter for pathogenic variants.

### RNA sequencing
RNA was extracted from samples collected from tumor, adjacent normal kidney and cell lines (Qiagen RNeasy or NEB Monarch Total RNA). For mechanistic studies, RNA was collected in biological replicates from cells

treated with shRNAs or compounds as listed in the figures. Libraries were prepared using Illumina TruSeq. Samples were run with at least 50 million paired-end reads using Novoseq 6000 (Illumina). Fastq or BAM files were mapped and aligned using Illumina Dragen v3.7.5 to GrCh38[66] and GenCode36[67] on Amazon Web Services. Samples were then quantified using salmon through Illumina Dragen v3.7.5. Gene count files were converted to a counts matrix using tximport[68]. TARGET clustering analysis in Fig. 2a utilized ComBat-seq[69] to correct for batch effects between studies. The counts matrix was used as input into DESeq2 to evaluate differential gene expression[70]. Log2 fold change values were used as input into GSEA[42] to measure gene set enrichment. Normalized read counts matrices were used as input into UMAP[28] to visualize clustering between samples. UMAP analysis was performed on the 1000 most highly variable genes in Wilms tumor as previously identified[71]. Versions used: R v4.2.2; R Studio 2022.07.1 Build 554; tximport v1.26.1, sva (ComBat-seq) v3.46.0, DESeq2 v1.38.3, umap v0.2.10.0, ggplot v3.4.2, Hmisc v4.6-0 (R packages); fgsea_1.24.0.

### Loss of function screens—methodology and analyses
RNAi and CRISPR-Cas9 screens were performed as previously described[20–22]. Specifically, we transduced into noted cell lines the DCT v1.0 libraries: shRNA (CP1050) and sgRNA (CP0026) libraries from the Broad Institute Genetic Perturbation Platform (GPP) (http://www.broadinstitute.org/rnai/public/). In parallel, PEDS_0023_T, PEDS_0023_T2, and PEDS_0002T cells were transduced with Cas9 expression vector pLX311_Cas9[72]. These stable cells were subsequently transduced using the DCT v2.0 libraries and at a multiplicity of infection (MOI) between 0.3 and 0.6[32]. Screens were performed with a goal representation rate of >500 cells per shRNA or sgRNA. Cells were passaged every 5–7 days until days 25–29 following transduction. Genomic DNA was extracted from an early time point and at the end time point. Samples were sequenced as previously described, deconvoluted and analyzed using MAGeK[33].

### Cell viability assays
Cells were plated in a concentration of 2500 cells/well in a 96-well plate (Corning, 3903, or Greiner Bio-One, 655098) and were incubated overnight in F-media. After 24 h, F-media was aspirated and F-media with the indicated drug concentration was added. Cells were treated between the range of 0.007 µM to 50 µM of KPT-330 (Selleck, S7252) and doxorubicin (Selleck, S1208) for 72 h. Cell viability was measured using CellTiter-Glo (Promega, G7573). Luminescence measured with BioTek Synergy MX with BioTek Gen5 v1.11.5 or Perkin Elmer 2103 with EnVision Manager 1.14.3049.1193. Cell viability was normalized to the luminescence of the control sample, with luminescence values of zero serving as 0% viability and the average luminescence value of the technical replicates of the control sample serving as 100% viability. IC$_{50}$s of KPT-330 and doxorubicin were determined in each cell line by fitting the dose response curve using Graphpad Prism v9.3.0. Each experiment was repeated in at least two biological replicates.

### *XPO1* and *TRIP13* shRNA experiments
shRNA sequences were designed using the Broad Institute GPP portal (http://www.broadinstitute.org/rnai/public/). Oligos were obtained from

Integrated DNA Technologies (IDT). Oligos were annealed and ligated with pLKO.5 as previously described. Constructs were transfected using TransIT-LT1 (Mirus Bio LLC, Madison, WI, USA) and HEK 293T cells. Cells were then transduced with lentivirus to achieve appropriate knockdown without viral toxicity as previously described[21]. Following selection with puromycin (Invivogen), cell proliferation was assayed by CellTiter-Glo at 10 days. Experiments were repeated a minimum of three times in technical triplicates. shRNA sequences can be found in Supplementary Table 4.

## Cell cycle analysis with flow cytometry
Cells were grown to 70–80% confluency and were treated or transduced as indicated in the figures. After treatment, cells were harvested and washed with cold PBS before being fixed in 70% ethanol for two hours on ice. Cells were stained with PI/RNase solution (Invitrogen, F10797) for fifteen minutes before being analyzed via flow cytometry. Data was collected with Beckman Coulter CytExpert v2.3. Data was analyzed in FlowJo v10.8.1. First, debris was removed via gating under FSC-A/SSC-A, singlets were gated with FSC-A/FSC-H, and then cell cycle phases were determined using the Watson (Pragmatic) model (Supplementary Fig. 7).

## TP53 CRISPR-Cas9 deletion
PEDS1012T and PEDS_0022T cells were transduced with Cas9 expression vector pLX311_Cas9. These cells were subsequently transduced with sgTP53-1 and sgLacZ in the pXPR003 backbone. Transduced cells were then cultured in the indicated antibiotics for selection and suppression of gene expression was confirmed by immunoblotting.

## qRT-PCR
Total RNA was extracted using the NEB Monarch RNA extraction kit. One microgram of RNA and oligo primers were used for cDNA synthesis in a total reaction volume of 20 μL High-Capacity cDNA Reverse Transcription Kit (ThermoFisher Scientific, 4368814). qRT-PCR reactions were prepared using SYBR-Green PCR Master Mix (ThermoFisher Scientific, 4367659) and run on a BioRad CFX96 qPCR System/BioRad CFX Manager v3.1 with a minimum of technical duplicates. Relative mRNA levels were calculated using the 2-ΔΔCt method[73]. Results shown are representative of at least two biological replicates. Primer sequences can be found in Supplementary Table 4.

## Immunoblots
Cells were grown to 70–80% confluency and were treated or transduced as indicated in the figures. Thereafter, cells were lysed using 1× RIPA (Cell Signaling Technologies, 9806) with protease inhibitors (coMplete, Roche, 42484600) and phosphatase inhibitors (PhosSTOP, Roche, 04906837001). For experiments using nuclear and cytoplasmic fractions, fractions were extracted using NE-PER (ThermoFisher Scientific, 78835). Using 10% or 4–12% SDS-PAGE gels, gels were transferred onto PVDF (Millipore, IPFL00010) or nitrocellulose membranes (ThermoFisher Scientific, IB23001). Blots were then visualized using Odyssey Classic v3.0.30 (Licor, Lincoln, NE). Antibodies used in this study include: XPO1 (Santa Cruz; sc-5595), β-Actin (C-4) (Santa Cruz; sc-47778), β-Actin (Cell Signaling; 8457), p53 (Santa Cruz; sc-126), TRIP13 (Abcam; ab128171), α-Tubulin (Santa Cruz, sc-5286), α-Tubulin (Cell Signaling; 2144), Lamin A/C (Cell Signaling; 4777 or 2032), p21 (Cell Signaling; 29475), CCND1 (Santa Cruz; sc-8396). Results shown are representative of at least two biological replicates.

## Synergy experiments
WT cells were plated in 96-well plates at a concentration of 2500 cells/well. WT cells were treated with either doxorubicin (62.5–500 nM), vincristine (62.5–500 nM), or actinomycin (62.5–500 nM) in combination with KPT-330 (1.25–10 μM). Each plate included a 4-dose dilution of each drug alone or in combination and included DMSO controls. Following a 3-day incubation, we performed a cell viability assay using CellTiter-Glo. Relative cell viabilities were then calculated following the procedure listed previously in cell viability assays and analyzed with SynergyFinder[74] using two alternate drug–drug interaction models: zero interaction potency (ZIP)[75] and Bliss independence[76]. Experiments were performed in biological replicates or triplicates. Raw data for calculations can be found in Supplementary Data 8.

## In vivo experiments
Female NSG (NOD-scid IL2Rgamma null) mice, 6 weeks old were purchased from The Jackson Laboratory (Bar Harbor, ME). Animals were acclimated for at least 5 days before initiation of the study. The study was conducted at Dana-Farber Cancer Institute (DFCI) with the approval of the Institutional Animal Care and Use Committee in an AAALAC accredited vivarium.

NSG mice were implanted with CCLF_PEDS_0041_T2 tumor fragments from previously expanded tumors, subcutaneously in the hind-flank. Tumors were allowed to establish to 86– 163 mm$^3$ (average 114.3 mm$^3$) in size before randomization using Studylog software (San Francisco, CA) into various treatment groups with 8 mice/group as follows: vehicle control (0.6% Pluronic F-68 + 0.6% Plasdone PVP; oral gavage on MWF × 4 weeks), selinexor (purchased from Selleck Chemicals LLC, 15 mg/kg oral gavage on MWF × 4 weeks), doxorubicin (obtained from DFCI pharmacy, 5 mg/kg intravenous injection on days 2 and 9) and the combination of selinexor and doxorubicin. No statistical methods were used to pre-determine sample sizes, but our sample sizes are similar to those reported in previous publications[21,22,77]. Once the treatment was completed, tumors were monitored at least once a week. For single agent and combination agent efficacy studies, mice were dosed for 21 or 28 days and monitored daily. Drugs were then withdrawn, and tumors were monitored twice weekly until study termination on day 150. Tumor volumes were calculated using the following formula: (mm$^3$) = length × width × width × 0.5. Mice were immediately euthanized if the tumor volume exceeded 2000 mm$^3$ or if the tumors became necrotic or ulcerated. The compounds were well tolerated with less than 8% body weight loss. Data collection and analysis were not performed blind to the conditions of the experiments. GraphPad Prism v9.3.0 was used to calculate significance in the Kaplan–Meier curves. We have complied with all relevant ethical regulations for animal use.

## Statistics and reproducibility
Comparisons between two groups were statistically evaluated by the student's paired or unpaired $t$ test, where applicable. Significance for Kaplan–Meier curves was performed using the log-rank (Mantel–Cox) test. Statistical significance for tumor volume between groups was performed using a one-way ANOVA test. Differences were considered significant at $P < 0.05$. Statistical analysis of RNA sequencing data was performed using DESeq2. All genes that had an FDR < 0.05 and a log2FoldChange > |1| were considered significant. At least two independent experiments with at least two technical replicates were performed to support statistically analyzed findings.

## Reporting summary
Further information on research design is available in the Nature Portfolio Reporting Summary linked to this article.

## Data availability
Sequence data have been deposited at the European Genome-phenome Archive (EGA), which is hosted by the EBI and the CRG, under accession number EGAS00001007389. This study does not use custom code or mathematical algorithms. The uncropped immunoblotting images were exhibited in Supplementary Fig. 8. The source data of the graph figures are exhibited in Supplementary Data 8. Plasmids herein can be found at https://www.addgene.org/Andrew_Hong/. All other data is available from the corresponding author upon reasonable request.

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

## Acknowledgements

We thank the patients and their families for their participation. We thank the Hong, Spangle, and Hahn labs for their thoughtful comments and suggestions. Tissue samples were provided by the Children's Healthcare of Atlanta Pediatric Bio-Repository or Dana-Farber/Boston Children's Center for Cancer and Blood Disorders. Other investigators may have received specimens from the same subjects. Research reported in this publication was supported in part by the Pediatrics/Winship Flow Cytometry Core of Winship Cancer Institute of Emory University, Children's Healthcare of Atlanta and NIH/NCI under award number P30CA138292, by the Bioinformatics and Systems Biology Shared and the Biostatistics Shared Resource of Winship Cancer Institute of Emory University and NIH/NCI under award number P30CA138292. The content is solely the responsibility of the authors and does not necessarily represent the official views of the National Institutes of Health. Research reported in this publication was supported by the following: NIGMS T32GM008490 (G.W.C.), NCI F31CA278008 (G.W.C.), DOD HT9425-23-0609 (Y.S.), NIGMS T32GM007739 (W.J.K.), NCI K08 CA2555569-01 (A.J.M.), NIH/NCI P30CA138292 (M.R.), NCI R35 CA231958 (D.M.W.), NIH R35 CA210030 (K.S.), NCI U01 CA176058 (W.C.H.), DOD W81XWH1910281 (A.L.H.), ACS MRSG-18-202-01 (A.L.H.). CureSearch for Children's Cancer YIA (A.L.H.), Rally for Childhood Cancer Investigator Grants 21IN12 and 22IC37 (A.L.H.).

## Author contributions

K.M., W.C.H. and A.L.H. in the design of the study. K.M., G.W.C., B.P.L., W.C.H. and A.L.H. wrote the manuscript. K.M., B.P.L., G.W.C., Y.S., K.S., B.D., M.R., M.B. and A.L.H performed analyses of the sequencing data. K.M., B.P.L., Y.S., W.J.K., M.D., B.D.K., P.K., Y.Y.T. J.B. and A.L.H. generated the cancer cell lines. J.S., H.C.J., T.L. and K.C.G. oversaw the Aflac Solid Tumor Biorepository. S.M., C.C. and B.C. oversaw the DF/BCH Solid Tumor Biorepository. W.J.K., M.D., D.A., B.D.K., X.Y., G.C., D.R. and A.L.H. performed and/or analyzed the functional genomic screens. A.L.C.,

B.A.L., J.S., H.C.J., C.M., A.J.M., K.C.G., K.L., D.M.W. and P.C.G. generated the PDXs and performed in vivo studies. M.B., B.C., Y.Y.T., J.B., K.L., D.R., A.J.M., D.M.W., P.C.G., J.M.S., M.R., E.A.M., K.S., K.C.G., W.C.H. and A.L.H. supervised the studies. All authors discussed the results and implications and edited the manuscript. All authors also read and approved the final manuscript.

## Competing interests

The authors declare the following competing financial interests. K.M. is currently employed at PreludeDx. A.L.C. is currently an employee of AstraZeneca. G.C. is currently an employee of Johnson & Johnson. D.M.W. is currently an employee of Merck and has research support from Daiichi Sankyo, Abcuro, Verastem, Secura, and is on the Advisory Board or has equity in Ajax, Travera, AstraZeneca, Bantam. K.S. receives grant funding from Novartis and KronosBio, consults for and has stock options in Auron Therapeutics and has served as an advisor for KronosBio and AstraZeneca. D.E.R. receives research funding from members of the Functional Genomics Consortium (Abbvie, BMS, Jannsen, Merck, Vir), and is a director of Addgene, Inc. W.C.H. is a consultant for ThermoFisher, Solasta Ventures, MPM Capital, KSQ Therapeutics, Tyra Biosciences, Jubilant Therapeutics, RAPPTA Therapeutics, Function Oncology, Frontier Medicines, Hexagon Biosciences, Serinus Biosciences, Kestral Therapeutics and Calyx. The remaining authors declare no competing interests.
