## [Peer Review File · Communications Biology]

Reviewers' comments:

Reviewer #1 (Remarks to the Author):

The authors described the establishment and characterization of primary patient-derived WT cell cultures. They used some of these cultures for RNAi and CRISPR-Cas9 screenings and tested nuclear export inhibition by KPT-330 as a therapeutic option in WT. Treatment with KPT-330 in combination with doxorubicin was tested in-vitro in primary WT cells and an in-vivo xenograft model. The authors claimed XPO1 inhibition by KPT-330 as a potential therapeutic option in WT treatment.

The manuscript is difficult to read. All in all, many experiments were performed and lots of data are collected but have been analysed in a biased, arbitrary manor and the conclusions seem far-fetched. Arbitrary subsets of different cell cultures were used, and it is not clear on what basis the cultures were selected for each experiment. Often the combination of different cultures was changed within a topic and the data is presented for only a subgroup of cultures at a time. In addition, the manuscript was not carefully prepared and checked with regard to data labeling. Several times sample names are mixed up, numbers are wrong or the links to figures or supplementary material are not correct. Inconsistencies, questionable interpretations and lack of stringency lead to an impression of a paper that may not (fully) meet criteria of a coherent story.

Specific points of criticism are listed as they appear in the manuscript:

1. Culture methods are described insufficiently: Which media is used (no “F-media” is mentioned in the ref 23)? Which supplements are included in the media? How were the cells detached for passaging? How many times could each cell culture be passaged before getting senescent?

2. Fig.1: No details on mutations are given. Please list all mutations in detail, not only with gene name (position, aa change, ...) . Compare tumor and cell culture data to ensure tumor origin of primary cultures and show data (WGS, WES) for tumor as well as for cell culture samples.

3. Only CNVs are given for cultures in suppl table 1. Did the authors test for LOH or CNV? They claimed “LOH” in the text but showed CNV only. LOH and CNV should be strictly separated and both shown in a suppl table. E.g. there is no CNV at 11p15 to be expected – copy-neutral LOH or loss of imprinting at this region (IGF2) are important driver events in WT. Check for IGF2 methylation status if you want to make a statement on 11p15.

Is there a difference between “gain” and “amplification” (suppl table 1)? What is the meaning of HLAMP? Fill in “q/p region” for all samples.

4. The cited TARGET reference Ooms 2016 (ref 26) evaluated DAWT only, not FHWT that were analysed in this draft. Mutational landscape might be different. Better compare with Gadd 2017 (ref 28). Some of the mutations listed are not common in WT: There are no somatic MYC (MYCC), NOTCH or PAX5 mutations reported to date. Without the exact description of the mutations, no statement can be made

as to whether the presented collection really reflects WT.

5. Does figure 1b depict RNAseq of primary cell cultures or tumor samples? Please show both, transcriptomes of tumor samples and matched cell cultures as well as the normal kidney samples used in this study.

SIX2 and CITED1 are upregulated in blastemal and less differentiated WT cells (doi: 10.1016/j.jpedsurg.2012.03.034). Do the 5/7 (FHWT) and 2/3 (DAWT) primary cultures with high SIX2 and CITED1 expression show signs of blastema or self-renewal capacity? What is the histological composition of the tumors they originate from?

6. Primary WT cells and PDX-derived cultures from the same tumor are not independent. Therefore, CCLF_PEDS_0023_T1 and _T2 resp. CCLF-PEDS_0041_T1 and _T2 should not be counted as two different cell cultures each (Fig.2).

What are the differences between CCLF-PEDS41_T1 and _T2? There is only little overlap with respect to RNAi screening.

A third independent cell culture with common WT alterations (e.g. Aflac_2365, Aflac_2377, or PEDS_1012) would significantly increase the validity of the screenings performed.

7. The authors listed only 7 genes overlapping in RNAi and CRISPR screens – but claimed 8 in text and figure 2b.

Text and fig. 2b state culture PEDS_0002T was used for CRISPR-Cas9 screen but data for PEDS_0022T is shown in Fig S1. Just mislabeling? Give numbers instead of colour code in fig S1, S1d-f suggest Pearson's correlation is 1 in early passages of all 3 cultures.

8. Show XPO1 expression in tumor samples and corresponding cell cultures as well as in normal tissue and kidney cultures. Since RNAseq was performed on all samples, the data should be available.

9. Fig 2d: Of 3 normal kidney cultures corresponding tumor cells were available only for one patient. Was there no kidney material available for the other patients? If so, test more different normal kidney cells, as the single normal – tumor pair (PEDS_0023) suggests partial sensitivity also in kidney cells. Why data for PEDS_0023_T2 is not depicted in fig. 2d? Show dose response curves for all cultures used in supplement.

10. Fig. s2: Why cell viability is not 100% in shControl (fig. s2b)? Mislabeling of PEDS_0041 in fig s2d. Show WB analysis and qRT-PCR for all cell lines – or at least coherent cultures, not different subsets. Include normal kidney cells in these analyses.

11. Experimental procedure for RNAseq of DMSO vs. KPT-300 treated PEDS_0041 is not described sufficiently. Which concentration of KPT-300 was used? Treatment duration? PEDS_0041 is the least sensitive cell culture of all those tested.

12. RNAseq of KPT treated cells: Of 412 differentially expressed genes 58 were “NA” (supp table 2) –

don't count them. XPO1 is not differentially expressed in RNAseq data of PEDS_0041, while it was upregulated (highly significant) by qRT-PCR (fig S2d). Please comment on discrepancies. Why "cell line 145T" is stated in suppl table 3?

13. The method of cell cycle analysis of KPT-treated cells is not described in detail. KPT-330 concentration, treatment duration? Have the cells been synchronized for cell cycle analysis? Proportion of cells in different cell cycle phases sums up to > 100% for PEDS_0022T - this is impossible.

14. Use TP53 mutant cell line PEDS_0023T, or any other TP53 mutant WT culture derived from a DAWT (WIT49, 17.94) as positive control for TP53 depletion. Would authors suggest a different mechanism of action in TP53 mutant cases, as resistance to KPT-330 due to TP53 loss of function is claimed in the discussion? PEDS_0023 is still sensitive to KPT-330.

15. Selection of TRIP13 for further functional studies remains arbitrary. TRIP13 is not reduced on protein level upon KPT treatment in PEDS_0002, _0041, _0022, Aflac_2365 (fig 2e). Biallelic functional LOSS of TRIP13 is described as WT predisposition, not TRIP13 GAIN of function (ref 42). So please discuss, why TRIP13 repression by shRNA reduce cell viability and TRIP13 overexpression should increase cell viability (not significant in fig s3g?).

16. Why is only partial data shown on TRIP13 shRNA experiments in fig S3? There is no data for Aflac_2377 presented, different shRNAs were used, no WB is done for CyclinD1 and TP53 for PEDS_1012. Data look inconsistent.

17. Add lists of differentially expressed genes and GSEA data corresponding to fig. 3f-h as supplement.

18. Why is doxorubicin selected for synergy experiments? Because doxorubicin has serious side effects and trials are trying to eliminate doxorubicin from treatment regimens, a better tolerated chemotherapeutic agent such as actinomycin or vincristine would be a better choice for combination therapy.

19. Doxo + KPT treatment: Data shown in figures 4b and 2d are contradictory: Also IC50 is < 1 μ M KPT for Aflac_2365, PEDS_1012, _0002, _0022 in fig2d, cell viability is 50% when treated with 5 μ M KPT in fig. 4b.

For synergy experiments samples shown were selected: PEDS_0023N is used for 4b, but not included in 4d, while PEDS_0040N, Aflac_2494N and Aflac_2597N are depicted in 4d only. The two Aflac normal kidney cells have never been mentioned before and there is no data available.

20. Why KPN1B is mentioned in abstract? There are no additional details on the impact of these findings and the effect of targeting KPN1B is not analysed further.

A similar study was presented by another group but has not yet been published: DOI:

10.1200/JCO.2020.38.15_suppl.3580 Journal of Clinical Oncology 38, no. 15_suppl (May 20, 2020) 3580-

3580.

Reviewer #2 (Remarks to the Author):

Nice and thorough report evaluating a preclinical model of WT, including one line having UH, for treatment sensitivity. This reviewer finds significant merit to publish the work for multiple purposes but principally the concept to establish and utilize WT cell cultures for rapid assessment of treatment sensitivity/resistance (ie, personalizing cancer care). Question arises whether validating these cultured lines/models retains the identical histologic elements of the primary tumor. Certainly the genetic composition appears retained for the most part but the reader is curious whether injection of these cell lines into the murine model recapitulates those of the standard PDX model described in the text: did the authors attempt to inject cultured cells and compare resulting in vivo tumors with the PDX method and then the primary WT? Then the investigator would have confidence with histologic retention and any genetics of resulting experimental tumors to conclude/establish rigorously the study observations. The question is begged whether certain cell types are selected in the culture conditions that are no longer representative of the primary tumor and then hence responsiveness to the in vitro drugs, which is a significant potential limitation of this approach. Overall this reviewer finds merit with the original manuscript and would be honored to review a revised version.

Reviewer #3 (Remarks to the Author):

Mittal et al. establish 10 patient-derived short-term Wilms tumor cell lines for in vitro and in vivo studies aimed at exploring novel treatment options. By combining RNAi and CRISPR/Cas9 loss-of-function screens with next generation sequencing, they identify the nuclear export gene XPO1 as a potential new target for inhibition with the FDA approved compound selinexor (KPT-330). KPT-330 effectively reduces the survival of Wilms tumors cells in the presence of P53. The study also provides functional insights showing that nuclear export inhibition causes G2/M arrest of the tumor-derived cell cultures. Furthermore, KPT-330 suppresses TRIP1, which is required for tumor cell survival. KPT-330 and doxorubicin synergistically increase the survival probability of mice xenotransplanted with patient-derived Wilms tumor cells.

In my opinion, this is a very interesting study of high quality. The work is technically sound and the reported data are original. Beyond that, the manuscript is very well written. I have a few comments and questions that should be addressed:

1. Fig. 1a: Inactivation of WTX at Xq11.2 has been reported in 18% to 30 % of Wilms tumors (Science 315: 642-645, 2007; Genes Chromosomes Cancer 47: 461-470, 2008). Furthermore, stimulation of the Wnt4/ β -catenin pathway due to activating CTNNB1 mutations occurs particularly in WT1-mutant nephroblastoma (Am. J. Pathol. 165: 1943-1953, 2004). You may thus consider providing information about the status of these genes in your tumor cell lines.

2. Fig. 1c: The transcriptome of tumor cells may change during ex vivo culture. What passage were the cultures at the time of RNA sequencing? Furthermore, did you characterize the normal kidney cells that were used as controls?
3. Fig. 1c, d: „MRT“ and “CCSK” should be defined. This can be done, for example, by adding the abbreviations to the corresponding legends in panel b of this figure.
4. Fig. 2b: What was your rationale for selecting these particular samples from your set of Wilms tumor cell cultures? Why did you use different lines (PEDS_0041 and PEDS_0023) for RNAi and CRISPR/Cas9 screening?
5. Fig. 2d: Obviously, the Wilms tumor-derived cells are more susceptible to KPT-330 in terms of viability than normal kidney cells. Is this simply due to the fact that they contain high levels of XPO1, i.e. does their susceptibility to KPT-330 correlate with XPO1 expression, or are there any other reasons underlying this phenomenon?
6. Fig. 2e: Upregulation of P53 in KPT-330-treated Wilms tumor cell lines is striking. Do you see a similar P53 increase also in normal kidney cells? The P53 immunoblot was performed with nuclear cell lysates, I guess. This should be mentioned in the figure legend.
7. Lines 139-142: „On target activity of KPT-330 requires transcriptional upregulation of XPO1 due to an auto-feedback loop in conjunction with suppression of XPO1 protein levels.” The mechanism of action of KPT-330 seems rather puzzling. Perhaps, you can provide a bit more information on it.
8. Fig. 3d: Deletion of TP53 increases IC50 values of KPT-330 between 6- and 13-fold suggesting that TP53 is required for KPT-330-induced cell death. However, this observation is somewhat in conflict with the low IC50 value of PEDS_0023_T, which harbors a TP53 mutation.
9. Line 234: “...anaplastic Wilms tumor...”

Reviewers' comments:

Reviewer #1 (Remarks to the Author):

The authors described the establishment and characterization of primary patient-derived WT cell cultures. They used some of these cultures for RNAi and CRISPR-Cas9 screenings and tested nuclear export inhibition by KPT-330 as a therapeutic option in WT. Treatment with KPT-330 in combination with doxorubicin was tested in-vitro in primary WT cells and an in-vivo xenograft model. The authors claimed XPO1 inhibition by KPT-330 as a potential therapeutic option in WT treatment.

We thank Reviewer #1 for identifying areas of improvement in our manuscript. We have since significantly revised the manuscript to address their concerns along with comments from Reviewers #2 and #3. Specifically, we have revised the manuscript to incorporate data from additional patient derived cell lines throughout our experiments.

Specific points of criticism are listed as they appear in the manuscript:

1. Culture methods are described insufficiently: Which media is used (no “F-media” is mentioned in the ref 23)? Which supplements are included in the media? How were the cells detached for passaging? How many times could each cell culture be passaged before getting senescent?

We have updated our Methods section to better characterize the culturing of our normal and WT cell lines (e.g., media and supplements) in **Supp Table 2**. To detach our cells for passaging, we used TrypLE (Gibco). We have included details on the number of times cells could be passaged in **Supp Table 3**. Specifically we have added the following text [lines 327-337] in the **Methods** of the revised manuscript:

“Tumor tissue or adjacent normal tissue was finely minced into 1-2mm³ pieces and cell lines were established using F-media as previously described (**Supp Table 2**)²³. Cells were detached for passaging using TrypLE (Gibco, 12605036).”

“In general, Wilms Tumor cells were grown for up to 30 passages and normal kidney cells were grown for approximately 12-15 passages before senescing (**Supp Table 3**).”

2. Fig.1: No details on mutations are given. Please list all mutations in detail, not only with gene name (position, aa change, ...). Compare tumor and cell culture data to ensure tumor origin of primary cultures and show data (WGS, WES) for tumor as well as for cell culture samples.

We have updated **Fig 1 (shown below)** along with **Supp Table 1** to incorporate the mutations identified with OpenCravat. This includes comparisons of the Normal, Tumor and associated Cell Lines.

1

Furthermore, we have added the following text [lines 99-101] in the revised manuscript:

“We then assessed the mutational profiles of these tumors and cell lines and observed genetic similarity between patient tumor samples and matched cell lines (Fig 1; Supp Table 1).”

- Only CNVs are given for cultures in suppl table 1. Did the authors test for LOH or CNV? They claimed “LOH” in the text but showed CNV only. LOH and CNV should be strictly separated and both shown in a suppl table. E.g. there is no CNV at 11p15 to be expected – copy-neutral LOH or loss of imprinting at this region (IGF2) are important driver events in WT. Check for IGF2 methylation status if you want to make a statement on 11p15. Is there a difference between “gain” and “amplification” (suppl table 1)? What is the meaning of HLAMP? Fill in “q/p region” for all samples.

We thank the reviewer for identifying this omission of our CNV/LOH results. Analysis with TITAN and ichorCNA indeed assesses for LOH, copy neutral LOH and CNV¹. We have incorporated these findings into **Fig 1** as in **Point #2 above** and provided this data in **Supp Table 1**. With respects to IGF2 methylation, we concur and have removed the 11p15 field to avoid misinterpretation of **Fig 1** as we did not perform methylation studies.

Furthermore, we have added the following text [lines 346-347] in the revised manuscript:

“Copy number alterations (amplification, gain, deletion) were assessed with ichorCNA⁵⁶ and TITAN⁵⁷. Loss of heterozygosity was calculated using TITAN.”

4. The cited TARGET reference Ooms 2016 (ref 26) evaluated DAWT only, not FHWT that were analysed in this draft. Mutational landscape might be different. Better compare with Gadd 2017 (ref 28). Some of the mutations listed are not common in WT: There are no somatic MYC (MYCC), NOTCH or PAX5 mutations reported to date. Without the exact description of the mutations, no statement can be made as to whether the presented collection really reflects WT.

We thank the reviewer for identifying this incorrect reference and have since corrected it to genes identified in Gadd 2017. However, the chosen genes were based on recent studies of WT PDXs published by our co-authors in 2019² and a recent study of genes and CNVs identified in relapsed Favorable Histology Wilms Tumor patients³. These references have been since added to the manuscript.

Furthermore, we have added the following text [lines 101-103] in the revised manuscript:

“We found that the observed mutations reflected the spectrum of mutations seen in the WT samples profiled in the NCI Therapeutically Applicable Research to Generate Effective Treatments (TARGET)²⁶ as well as other studies”

5. Does figure 1b depict RNAseq of primary cell cultures or tumor samples? Please show both, transcriptomes of tumor samples and matched cell cultures as well as the normal kidney samples used in this study. SIX2 and CITED1 are upregulated in blastemal and less differentiated WT cells (doi: 10.1016/j.jpedsurg.2012.03.034). Do the 5/7 (FHWT) and 2/3 (DAWT) primary cultures with high SIX2 and CITED1 expression show signs of blastema or self-renewal capacity? What is the histological composition of the tumors they originate from?

Fig 1b (now Fig 2a) was initially depicting the primary cell cultures and tumor samples. However, the reviewer is correct that this could be further clarified. We have thus revised **Fig 2a** to more clearly depict the clustering of Wilms' tumor samples and matched cell cultures in comparison to other pediatric tumor samples.

Given the limited tissue obtained at time of procurement, we do not have formalin fixed tissue to perform H&E stains and determine the histological composition of our samples. From the pathology reports, triphasic biology was identified but discussion of blastemal predominance was not consistently captured.

We have added the following text [lines 109-110] in the revised manuscript:

“We observed that our WT cell lines clustered closely with the WT tumor samples and our normal tissue samples cluster closely with prior normal kidney samples (**Fig 2a**).”

6. Primary WT cells and PDX-derived cultures from the same tumor are not independent. Therefore, CCLF_PEDS_0023_T1 and _T2 resp. CCLF-PEDS_0041_T1 and _T2 should not be counted as two different cell cultures each (Fig.2). What are the differences between CCLF-PEDS41_T1 and _T2? There is only little overlap with respect to RNAi screening. A third independent cell culture with common WT alterations (e.g. Aflac_2365, Aflac_2377, or PEDS_1012) would significantly increase the validity of the screenings performed.

The differences between T1 and T2 are that T1 reflects the cell line generated from the primary tumor and T2 reflects the cell line generated from the PDX. We would note that given the heterogeneity of Wilms Tumor and cancers in general, that these may not be identical. For example PEDS_0023 has 1q gain in the cell line but not in the PDX. However in the case of PEDS_0041, we did not identify differences between the cell line and PDX. To address the concerns of the reviewer, we have modified our **Fig 2b** (now **Fig 3b**) to reflect unique patients and added **Supp Fig 1j and 1k** which shows the details of these two separate screens for each patient.

We agree that additional cell cultures which can tolerate high throughput genomic screens would be ideal. We have tried to expand our remaining WT cell lines (e.g., minimum of 100 million cells per screen

per biological replicate) but were unable to do so. Further, functional genomic screens require lentiviral titration studies, incorporation of Cas9 for CRISPR-Cas9 experiments, and ability to infect cells to achieve a multiplicity of infection of 20-60%⁴. Prior experiences through the Broad Genomics Perturbation Platform show that even robust cancer cell lines will fail screening for a multitude of reasons.

However, we have active efforts to identify cytokines and growth factors needed to support the continued expansion of our WT cell cultures with the hopes that we can perform genome-wide loss of function screens in the future.

We have added the following text [lines 118-123 and 314-319] in the revised manuscript to reflect this discussion:

“Following rapid expansion within the first five passages, we subjected cell lines derived from three unique patient samples (CCLF_PEDS_0002, CCLF_PEDS_0023, and CCLF_PEDS_0041) to targeted loss of function RNA interference and CRISPR-Cas9 screens (Methods, Supp Table 4). Given the limited number of cells that could be expanded, we used the Druggable Cancer Targets (DCT) library consisting of 429 genes that focused on known or upcoming therapeutic targets that had small molecule inhibitors^{23,32} (Fig 3a).”

“Finally, there are limitations to our studies. First, these studies utilized short term patient derived cell lines. Some features found in patient tumors are likely lost or enriched in the cultured environment, particularly at later passages. Future studies to dissect out the triphasic biology at single cell resolution are needed. In addition, limited expansion of these cell lines prevents high throughput studies across our cell lines. The use of immortalization techniques may alter our assessment of the biology of WT so identifying growth factors or cytokines to support in vitro growth is needed.

7. The authors listed only 7 genes overlapping in RNAi and CRISPR screens – but claimed 8 in text and figure 2b. Text and fig. 2b state culture PEDS_0002T was used for CRISPR-Cas9 screen but data for PEDS_0022T is shown in Fig S1. Just mislabeling? Give numbers instead of colour code in fig S1, S1d-f suggest Pearson’s correlation is 1 in early passages of all 3 cultures.

We thank the reviewer for catching this typographical error. We have since modified the text to correctly reflect the numbers and cell line names in the revised **Fig 3b**. We have further added numbers to the color code in **Fig S1a-i**.

We have revised the following text [lines 130-131] in the revised manuscript:

“From these orthogonal screens, we identified seven genes which overlapped between RNAi and CRISPR Cas9 screens.”

8. Show XPO1 expression in tumor samples and corresponding cell cultures as well as in normal tissue and kidney cultures. Since RNAseq was performed on all samples, the data should be available.

We have added the recommended **Fig 2e** to reflect *XPO1* expression in normal tissue, in normal kidney cultures, in tumor samples and corresponding tumor cell cultures.

9. Fig 2d: Of 3 normal kidney cultures corresponding tumor cells were available only for one patient. Was there no kidney material available for the other patients? If so, test more different normal kidney cells, as the single normal – tumor pair (PEDS_0023) suggests partial sensitivity also in kidney cells. Why data for PEDS_0023_T2 is not depicted in fig. 2d? Show dose response curves for all cultures used in supplement.

Obtaining adjacent normal kidney cells both requires IRB consent/assent as well as having tissue available following surgery. Furthermore, the take rate of growing normal kidney cells *in vitro* is limited in our experience. As such, we had limited numbers of matched normal cell lines in our initial submission. We have since been able to grow additional matched normal and tumor pairs from 2 patients: Aflac_2494 and Aflac_2597. We have further added in PEDS_0023_T2 in **Fig 3d (was Fig 2d previously)**. Finally, we have added the dose response curves for all cultures used in **Supp Figure 4**.

We have revised the following text [lines 142-149] in the revised manuscript:

“Specifically, we determined the IC₅₀ values for our WT cell lines and compared these to normal kidney cell lines. Among tumor cell lines we observed an average IC₅₀ of 1.79 uM ± 1.8, with over half of these cell lines showing an IC₅₀ at nanomolar concentrations (e.g. 25-800nM). Among normal cell lines we observed of an average IC₅₀ of 11.64uM ± 6.6 (**Fig 3d**). Moreover, we observed increased sensitivity to KPT-330 in all tumor cell lines for the three matched tumor-normal pairs (PEDS_023, Aflac_2494, Aflac_2597). The increased sensitivity of tumor cells as compared to normal cells suggests KPT-330 may be a selective inhibitor with limited off-target toxicity.”

10. Fig. s2: Why cell viability is not 100% in shControl (fig. s2b)? Mislabeling of PEDS_0041 in fig s2d. Show WB analysis and qRT-PCR for all cell lines – or at least coherent cultures, not different subsets. Include normal kidney cells in these analyses.

We thank the reviewer for identifying that in PEDS1012T, the shControl had a level above 100%. The reviewer is correct that when normalized it should be 100%. Specifically, we used two shRNA controls. One was used to normalize our results, and this was not the case in PEDS1012T. We have since corrected **Supp Fig 2b**.

Further, we have corrected the labeling of PEDS_0041.

We have added an immunoblot showing *XPO1* suppression in the 7 tumor cell lines (Supp Fig 2a) which were used for viability assays in Supp 2b.

We have further included an immunoblot following KPT treatment (5uM) for five normal cell lines, which show diminished P53 accumulation as compared to tumor cell lines (Supp Fig 2d).

We have further included qRT-PCR analysis in 7 tumor cell lines and 5 normal cell lines, measuring *XPO1* expression following treatment with KPT-330 for in both normal and tumor (**Supp Fig 2e**). We saw a significant increase in *XPO1* expression upon KPT-330 treatment in the tumor cell lines as compared to the normal cell lines.

11. Experimental procedure for RNAseq of DMSO vs. KPT-300 treated PEDS_0041 is not described sufficiently. Which concentration of KPT-300 was used? Treatment duration? PEDS_0041 is the least sensitive cell culture of all those tested.

We have incorporated additional details into the experimental procedures of how the RNA-sequencing was performed into the Methods section. We have added the following text [lines 364-369] in the revised manuscript:

“Gene count files were converted to a counts matrix using tximport. TARGET clustering analysis in **Fig 2a** utilized ComBat-seq to correct for batch effects between studies. The counts matrix was used as input into DESeq2 to evaluate differential gene expression. Log2 fold change values were used as input into GSEA to measure gene set enrichment. Normalized read counts matrices were used as input into UMAP to visualize clustering between samples.”

Specifically, we treated cells with 6uM KPT-330, the corresponding IC50 for PEDS_0041, for 24 hours. We have added the following text [lines 164-166] in the revised manuscript:

“To understand the transcriptional changes driving this G2/M arrest, we treated TP53 wild-type CCLF_PEDS_0041_T1 with DMSO or KPT-330 (6 uM) for 24 hours and performed RNA sequencing (**Methods**).”

12. RNAseq of KPT treated cells: Of 412 differentially expressed genes 58 were “NA” (supp table 2) – don’t count them. *XPO1* is not differentially expressed in RNAseq data of PEDS_0041, while it was upregulated (highly significant) by qRT-PCR (fig S2d). Please comment on discrepancies. Why “cell line 145T” is stated in suppl table 3?

We thank the reviewer for this comment. The “NA” were genes which did not have a defined name but had a gene (ENSG) identifier. We have since removed these per the reviewer’s request.

We have since standardized our analyses across this manuscript to use DESeq2⁵ and revised our volcano plots to show all genes with a p-value <0.05 and log2 fold change > 1 or < -1 as significant. In our revised manuscript, *XPO1* is differentially expressed with a log2 fold change of 1.02 which is consistent with our qRT-PCR data (**Supp Fig 4b**). We have updated these results into the **Supp Table 6**.

We thank the reviewer for noting the notes of “cell line 145T.” We apologize for this column as the original cell line nomenclature was Peds145T but was formally named PEDS_0041_T1 for this manuscript.

13. The method of cell cycle analysis of KPT-treated cells is not described in detail. KPT-330 concentration, treatment duration? Have the cells been synchronized for cell cycle analysis? Proportion of cells in different cell cycle phases sums up to > 100% for PEDS_0022T - this is impossible.

We have included the details of the cell cycle analyses into the methods of the manuscript (5uM KPT-330 for 24 hours). This includes the fact that the cells were not synchronized for cell cycle analyses. We have since revised the figure to include cells in <G1, G1, S, G2/M, and >G2. and provided the percentages from the biological replicates in **Supp Table 5**.

14. Use TP53 mutant cell line PEDS_0023T, or any other TP53 mutant WT culture derived from a DAWT (WiT49, 17.94) as positive control for TP53 depletion. Would authors suggest a different mechanism of action in TP53 mutant cases, as resistance to KPT-330 due to TP53 loss of function is claimed in the discussion? PEDS_0023 is still sensitive to KPT-330.

This manuscript shows that genetic ablation of TP53 in two favorable histology Wilms tumor cultures, PEDS1012T and PEDS_0022T, leads to decreased sensitivity to KPT-330. To assess the role of TP53 in the KPT-330 response in the context of DAWT, we attempted to delete *TP53* in PEDS_0023T using Cas9 mediated cleavage using the same sgRNAs used in PEDS1012T and PEDS_0022T. After multiple attempts, we were unable to generate a stable cell line in PEDS_0023T1 where *TP53* was deleted. We looked at the *TP53* mutation as identified in our analyses for **Fig 1** and we observed PEDS_0023T1 (as well as the primary tumor and the PDX) have a homozygous c.469G>T (p.Val157Phe). Previous studies suggest this alteration may confer an oncogenic gain of function TP53⁶. Thus cells are likely addicted to this mutant TP53 and deletion is not feasible. The study of TP53 mutations in DAWT remains limited. Further only 48% of DAWT present with a TP53 mutation, suggesting that TP53 mutations are not the only factor driving anaplasia⁷. There is ongoing work within the lab to better understand the effect of KPT-330 treatment on TP53 mutant WT cell lines.

We have added the following text [lines 304-312] in the revised manuscript to address this:

“We have observed that the DAWT models used in this study, PEDS_0023_T1 and T2, are sensitive to nuclear export inhibition. Sequencing analysis revealed that these models have a putative oncogenic-gain-of-function TP53 mutation (p.Val157Phe) (Fig 1; Supp Table 1)⁶¹. To discern if nuclear mutant TP53

accumulation is driving this response in DAWT, we attempted to delete TP53 using CRISPR-Cas9 similar to PEDS1012T and PEDS_0022_T. We observed that these models were unable to tolerate deletion of TP53, suggesting that these cells rely on mutant TP53 for their proliferation. Together these observations suggest that PEDS_0023_T may have another mechanism through which nuclear export inhibition functions. Further studies with additional models are needed to fully characterize the molecular mechanisms driving KPT-330 activity in DAWT.”

15. Selection of TRIP13 for further functional studies remains arbitrary. TRIP13 is not reduced on protein level upon KPT treatment in PEDS_0002, _0041, _0022, Aflac_2365 (fig 2e).

We appreciate that our explanation of how we arrived at TRIP13 for further studies was not clear. We have since re-written this portion of the manuscript to clarify our rationale. We have added the following lines of text [lines 159-169] in the revised manuscript:

“We performed fluorescence-activated single cell analyses to assess the changes in the cell cycle following KPT-330 treatment. We found that changes in G1 were not consistent across our FHWT cell lines (Fig 4a). However, we saw decreases of S phase and significant increases in G2/M suggesting that KPT-330 in our FHWT cell lines led primarily to a G2/M arrest (Fig 4a). We focused our efforts on understanding the mechanisms in FHWT. To understand the transcriptional changes driving this G2/M arrest, we treated TP53 wild-type CCLF_PEDS_0041T1 with DMSO or KPT-330 using the IC50 concentrations (e.g., 6 μ M) for 24 hours and performed RNA-sequencing (Methods). We then performed differential expression analyses and found 1,120 genes differentially expressed (Fig 4b, Supp Table 6). We examined gene sets enriched or suppressed upon KPT-330 treatment using Gene Set Enrichment Analyses (GSEA)⁴³ and observed 32 hallmark gene sets significantly enriched or suppressed (Supp Table 7).”

With regards to our protein levels, we have since repeated KPT treatment in four of the previous cell lines and have observed a more robust decrease in TRIP13 levels (likely due to increase protein input and further antibody optimization).

Biallelic functional LOSS of TRIP13 is described as WT predisposition, not TRIP13 GAIN of function (ref 42). So please discuss, why TRIP13 repression by shRNA reduce cell viability and TRIP13 overexpression should increase cell viability (not significant in fig s3g?).

We have completed several biological replicates of our TRIP13 lentiviral overexpressed cells in **Figure S5h** and found that there is a modest increase in viability when compared to a Luciferase overexpression control. Specifically, it is significant in PEDS_1012T whereas it did not achieve significance in PEDS_0022T. This would suggest that in the context of cancer cell maintenance, that gain of TRIP13

does not lead to decreased cell viability. As Reviewer #1 noted, this is different than findings seen in ref 42. However, ref 42 used HEK293Ts to perform their TRIP13 overexpression studies whereas we are using Wilms tumor cell lines to assess the role of TRIP13. It is possible that TRIP13 overexpression has differing roles in different lineages and contexts.

We have added the following text [lines 197-199] in the revised manuscript:

“Previous observations in HEK293T have reported a decrease in proliferation upon TRIP13 overexpression suggesting TRIP13 has differing roles in different lineages and contexts.”

16. Why is only partial data shown on TRIP13 shRNA experiments in fig S3? There is no data for Aflac_2377 presented, different shRNAs were used, no WB is done for CyclinD1 and TP53 for PEDS_1012. Data look inconsistent.

We apologize to the reviewer for this omission from our initial submission. We have since included Aflac_2377 and repeated our immunoblots probing for Cyclin D1 and TP53 which are now shown in **Supp Figs 5e and g**.

17. Add lists of differentially expressed genes and GSEA data corresponding to fig. 3f-h as supplement.

Per request of the reviewer, we have added the differentially expressed genes and GSEA data corresponding to revised **Figure 4f-h** in **Supp Tables 8 (shTRIP13 diff exp genes)** and **9 (GSEA)**.

18. Why is doxorubicin selected for synergy experiments? Because doxorubicin has serious side effects and trials are trying to eliminate doxorubicin from treatment regimens, a better tolerated chemotherapeutic agent such as actinomycin or vincristine would be a better choice for combination therapy.

Reviewer #1 is correct that the risks for cardiac toxicities are elevated in patients treated with doxorubicin. However in the Children's Oncology Group AREN0533 trial, patients with Stage IV FHWT who had intensification of therapy (which increased cumulative doxorubicin doses of 150mg/m² with DD4A to 195mg/m² with Regimen M) led to overall improved survival, particularly those with rapid pulmonary nodule responses⁸. Furthermore, of patients with high-risk Favorable Histology Wilms whose cancers recur (~20% of patients), only 50% are salvaged despite further intensification of therapy which also includes additional doxorubicin (COG AREN1921). We therefore opted to determine if KPT-330 could synergize with doxorubicin with one of our least sensitive patient cell lines, PEDS_0041T, as this is an area of unmet need. Indeed we found that doxorubicin alone was unable to sufficiently lead to tumor regression and that addition of KPT-330 led to durable regression in 5 of 8 mice (**revised Fig 5d-f**).

To address Reviewer #1's concerns, we further assessed in two FHWT cell lines if either vincristine or actinomycin were synergistic with KPT-330. We observed a non-synergistic relationship for both vincristine and dactinomycin over two biological replicates (**Fig 5a**).

We have further added the following text [lines 240-243] in the revised manuscript:

"We assessed the role of adding KPT-330 to vincristine, dactinomycin or doxorubicin to determine if there was synergy or additivity with our cell line models of FHWT as measured by CellTiter-Glo. We found the combination of KPT-330 with vincristine or actinomycin with KPT-330 was not synergistic whereas KPT-330 had a promising additive to synergistic effect with doxorubicin (Fig 5a)."

19. Doxo + KPT treatment: Data shown in figures 4b and 2d are contradictory: Also IC₅₀ is < 1 μM KPT for Aflac_2365, PEDS_1012, _0002, _0022 in fig2d, cell viability is 50% when treated with 5 μM KPT in fig. 4b.

We thank the reviewer for identifying this. We agree with the reviewer this data was confusing. As such, we have removed sub-panel 4b.

For synergy experiments samples shown were selected: PEDS_0023N is used for 4b, but not included in 4d, while PEDS_0040N, Aflac_2494N and Aflac_2597N are depicted in 4d only. The two Aflac normal kidney cells have never been mentioned before and there is no data available.

We thank the reviewer for this question. We have since included all 12 tumor cell lines and all 5 normal cell lines in Fig 1, Fig 3d, Fig 5b, and Fig 5c. We have also since repeated the synergy experiments shown in revised Fig 5c as well as added 2 matched normal/tumor pairs (Aflac_2494 and Aflac_2597) per Reviewer #1 Question #9.

We have added the following text [lines 247-254] in the revised manuscript:

“Subsequently, we evaluated the effect of combination KPT-330 and doxorubicin treatment across our cell lines. The combination was synergistic (e.g., scores >10) in 25% of cell lines tested (3 out of 12) across multiple concentrations of KPT-330 and doxorubicin (Fig. 5c). For the remaining 9 tumor cell lines, two of which were DAWT, the combination was found to be additive with synergy scores ranging from 1 to 9. In contrast, this combination was not synergistic in all the normal cell lines with scores ranging from -5 to -12 (Fig. 5c).”

20. Why KPN1B is mentioned in abstract? There are no additional details on the impact of these findings and the effect of targeting KPN1B is not analyzed further.

We have removed this per the reviewers' request.

21. A similar study was presented by another group but has not yet been published: DOI: 10.1200/JCO.2020.38.15_suppl.3580 Journal of Clinical Oncology 38, no. 15_suppl (May 20, 2020) 3580-3580.

We thank the reviewer for identifying this abstract. We have been in contact with the Cruz and Kung labs since our initial submission about these efforts. We were excited to see that their work looking at XPO1 inhibition as monotherapies being recently published⁹. Our work complements their findings as well as looks at the role of combination therapy with a commonly used chemotherapy in FHWT treatment, doxorubicin. Furthermore, our study provides additional mechanistic insights. As this article has now been published, we have incorporated the reference into the manuscript.

We have added the following text [lines 136-137] in the revised manuscript:

“Further, KPT-330 has recently been identified as a potential effective therapy which targets renal tumors with aberrant XPO1 activation.”

Reviewer #2 (Remarks to the Author):

Nice and thorough report evaluating a preclinical model of WT, including one line having UH, for treatment sensitivity. This reviewer finds significant merit to publish the work for multiple purposes but principally the concept to establish and utilize WT cell cultures for rapid assessment of treatment sensitivity/resistance (ie, personalizing cancer care). Question arises whether validating these cultured lines/models retains the identical histologic elements of the primary tumor. Certainly the genetic composition appears retained for the most part but the reader is curious whether injection of these cell lines into the murine model recapitulates those of the standard PDX model described in the text: did the authors attempt to inject cultured cells and compare resulting in vivo tumors with the PDX method and then the primary WT? Then the investigator would have confidence with histologic retention and any genetics of resulting experimental tumors to conclude/establish rigorously the study observations. The question is begged whether certain cell types are selected in the culture conditions that are

no longer representative of the primary tumor and then hence responsiveness to the *in vitro* drugs, which is a significant potential limitation of this approach. Overall this reviewer finds merit with the original manuscript and would be honored to review a revised version.

We thank the reviewer for their comments on our manuscript. We appreciate the thoughtful feedback. To address this question, we took the PEDS_0041 cell line derived from the patient and injected 5 million and 10 million cells into the flanks of NOD.Cg-Prkdc^{scid} Il2rg^{tm1Wjl}/SzJ (Strain #:005557) mice subcutaneously that mixed in Matrigel. Following several months, unfortunately we failed to see tumor growth *in vivo*.

In parallel, we performed RNA-sequencing at single cell resolution in Aflac2597T from the primary tumor and derived cell line. We preliminarily found that our cell line retains many of the components at an early passage but we also see signs of differentiation through expansion of the UBCD (ureteric bud/collecting duct) population. Future studies in our lab are looking to better understand this process as we were unsuccessful in generating a PDX from this patient. Cell types were identified using markers identified in previous publication looking at kidney tumors¹⁰.

We have added the following text [lines 313-318] in the revised manuscript to reflect limitations of this work:

“Finally, there are limitations to our studies. First, these studies utilized short term patient derived cell lines. Some features found in patient tumors are likely lost in the cultured environment, particularly at later passages. Future studies to dissect out the triphasic biology at single cell resolution are needed. In addition, limited expansion of these cell lines prevents high throughput studies across our cell lines. The use of immortalization techniques may alter our assessment of the biology of WT so identifying growth factors or cytokines to support *in vitro* growth is needed.”

Reviewer #3 (Remarks to the Author):

Mittal et al. establish 10 patient-derived short-term Wilms tumor cell lines for *in vitro* and *in vivo* studies aimed at exploring novel treatment options. By combining RNAi and CRISPR/Cas9 loss-of-function screens with next generation sequencing, they identify the nuclear export gene XPO1 as a potential new target for inhibition with the FDA approved compound selinexor (KPT-330). KPT-330 effectively reduces the survival of Wilms tumor cells in the presence of P53. The study also provides functional insights showing that nuclear export inhibition causes G2/M arrest of the tumor-derived cell cultures. Furthermore, KPT-330 suppresses TRIP1, which is

required for tumor cell survival. KPT-330 and doxorubicin synergistically increase the survival probability of mice xenotransplanted with patient-derived Wilms tumor cells.

In my opinion, this is a very interesting study of high quality. The work is technically sound and the reported data are original.

We thank the reviewer for the comments on our manuscript. We appreciate the thoughtful feedback.

Beyond that, the manuscript is very well written. I have a few comments and questions that should be addressed:

1. Fig. 1a: Inactivation of WTX at Xq11.2 has been reported in 18% to 30 % of Wilms tumors (Science 315: 642-645, 2007; Genes Chromosomes Cancer 47: 461-470, 2008). Furthermore, stimulation of the Wnt4/ β -catenin pathway due to activating CTNNB1 mutations occurs particularly in WT1-mutant nephroblastoma (Am. J. Pathol. 165: 1943-1953, 2004). You may thus consider providing information about the status of these genes in your tumor cell lines.

We thank the reviewer for this suggestion. We have reanalyzed our data to look for Xq11.2 and Wnt4/ β -catenin pathway gene alterations. We observed a CTNNB1 mutation in 1 out of 10 patient samples (Aflac_2494T), which co-occurred with a WT1 mutation (**Figure 1**). However we did not observe mutations in WTX in these patient samples, which has also been added to **Figure 1**.

2. Fig. 1c: The transcriptome of tumor cells may change during ex vivo culture. What passage were the cultures at the time of RNA sequencing? Furthermore, did you characterize the normal kidney cells that were used as controls?

We thank the reviewer for raising this point. We agree that the transcriptome of tumor (and normal) cells may change during *ex vivo* culture especially as these cells will slow down in growth over time. We have now included the passage numbers at time of RNA sequencing for each of cell lines used in this study in relation to the primary tissue sample (**Supp Table 3**). These were performed at early and mid-timepoints which reflect when our studies were performed. We also performed transcriptomics at single cell resolution and saw that these retain most features at the transcriptome but also recognize the increase in the Ureteric Bud/Collecting Duct (UBCD) cell population (**See Reviewer #2, Comment #1**). Future studies are aimed at looking at these changes.

As for characterization of the normal kidney cells, in our revised manuscript, we have since added DNA and RNA-sequencing from our patients where tissue and/or cell lines were available. We have incorporated these results into revised **Figures 1 and 2**. We have furthermore added 2 matched normal cell lines/tumor cell lines based on feedback from Reviewer #1. We have further integrated these cell lines into the remainder of the revised manuscript. Specifically, we added IC50s for KPT330 and doxorubicin (**Figure 3d, 5b**), synergy analyses (**Figure 5c**), protein levels of XPO1 and TP53 in **Supp Fig 2d**.

We have added the following text [lines 109-110] in the revised manuscript:

“We observed that our WT cell lines clustered closely with the WT tumor samples and our normal tissue samples cluster closely with prior normal kidney samples (**Fig 2a**).”

3. Fig. 1c, d: „MRT“ and “CCSK” should be defined. This can be done, for example, by adding the abbreviations to the corresponding legends in panel b of this figure.

We thank the reviewer for identifying this. We have clarified this in revised **Figure 2**.

4. Fig. 2b: What was your rationale for selecting these particular samples from your set of Wilms tumor cell cultures? Why did you use different lines (PEDS_0041 and PEDS_0023) for RNAi and CRISPR/Cas9 screening?

Our rationale was based on technical feasibility as described in the response to Reviewer #1 Question #6. In brief, to perform such experiments we require at least 100 million cells at each passage which can be passaged for 21-35 days. We had tried performing this in additional cell lines but were not successful. We have current efforts to identify new modalities of expanding Wilms tumor cultures without the need to genetically alter the cell lines (e.g., through alterations in TP53 or TERT).

5. Fig. 2d: Obviously, the Wilms tumor-derived cells are more susceptible to KPT-330 in terms of viability than normal kidney cells. Is this simply due to the fact that they contain high levels of XPO1, i.e. does their susceptibility to KPT-330 correlate with XPO1 expression, or are there any other reasons underlying this phenomenon?

The reviewer is correct that *XPO1* levels are elevated in patients with WT or our WT cell lines (revised **Figures 2e** and **3c**). We thank the reviewer for the idea to assess correlation between *XPO1* levels and sensitivity to KPT-330. We looked at the statistical relationship between KPT-330 susceptibility *XPO1* levels and found a low R² of 0.06125 **Supp Fig 4a**. This suggests that susceptibility is not directly associated with expression levels of *XPO1*.

We have added the following text [lines 149-152] in the revised manuscript:

“We observed that *XPO1* expression was not correlated with KPT-330 sensitivity ($r^2 = 0.06125$, Supp Fig 4a) suggesting that on target activity is not entirely dependent on elevated *XPO1* transcript levels. However, we found that the relative fold change in *XPO1* levels was significantly higher in tumor cell lines as compared to normal cell lines (p-value 0.044; Supp Fig 4b).”

6. Fig. 2e: Upregulation of P53 in KPT-330-treated Wilms tumor cell lines is striking. Do you see a similar P53 increase also in normal kidney cells? The P53 immunoblot was performed with nuclear cell lysates, I guess. This should be mentioned in the figure legend.

We assessed whether treatment of normal cell lines (PEDS_0023N, PEDS_040N, PEDS_066N, Aflac_2494N, Aflac_2597N) with KPT-330 increased TP53 levels as they do in Wilms tumor cell lines (revised **Supplementary Figure 4d**). We found that there was a modest increase in TP53 levels upon KPT-330 treatment in normal cell lines as compared to the Wilms' tumor cell lines.

Of note, our findings were based on total lysate rather than nuclear cell lysate. We have since clarified this in the text and figure legend.

We have added the following text [lines 727-729] in the revised manuscript:

“(e) Immunoblots depicting the decrease in total protein levels of *XPO1*, TRIP13, and TRIP13 upon treatment with KPT-330.”

7. Lines 139-142: „On target activity of KPT-330 requires transcriptional upregulation of *XPO1* due to an auto-feedback loop in conjunction with suppression of *XPO1* protein levels.” The mechanism of action of KPT-330 seems rather puzzling. Perhaps, you can provide a bit more information on it.

We thank the reviewer for identifying this section of text that is not clear. We have since edited the text and added additional references to provide clarity on the known mechanisms of KPT-330 at the transcriptional level.

We have added the following text in the manuscript [Lines 153-155]

“These findings are consistent with prior findings that suggest on target activity of KPT-330 decreases the abundance of XPO1 protein, inducing a positive feedback loop which increases XPO1 mRNA levels.”

8. Fig. 3d: Deletion of TP53 increases IC50 values of KPT-330 between 6- and 13-fold suggesting that TP53 is required for KPT-330-induced cell death. However, this observation is somewhat in conflict with the low IC50 value of PEDS_0023_T, which harbors a TP53 mutation.

We thank the reviewer for this question which is similar to that of Reviewer #1 Question #14. Briefly, PEDS_0023_T has a homozygous TP53 c.469G>T (p.Val157Phe) mutation. This mutation has been characterized as an oncogenic gain-of-function mutation⁶. We assessed if depletion of TP53 in PEDS_0023_T affects sensitivity to KPT-330 using CRISPR-Cas9. We observed that cells were unable to survive upon TP53 depletion, suggesting that this cell line relies on mutant TP53 for its proliferation. This suggests that the type of TP53 mutation is important in DAWT and future studies will better understand these differences.

We have added the following text [lines 304-213] in the revised manuscript:

“We have observed that the DAWT models used in this study, PEDS_0023_T1 and T2, are sensitive to nuclear export inhibition. Sequencing analysis revealed that these models have a putative oncogenic-gain-of-function TP53 mutation (p.Val157Phe) (Fig 1; Supp Table 1)⁶. To discern if nuclear mutant TP53 accumulation is driving this response in DAWT, we attempted to delete TP53 using CRISPR-Cas9 similar to PEDS1012T and PEDS_0022_T. We observed that these models were unable to tolerate deletion of TP53, suggesting that these cells rely on mutant TP53 for their proliferation. Together these observations suggest that PEDS_0023_T may have another mechanism through which nuclear export inhibition functions. Further studies with additional models are needed to fully characterize the molecular mechanisms driving KPT-330 activity in DAWT.”

9. Line 234: “...anaplastic Wilms tumor...”

We thank the reviewer for identifying this typographical omission. We have modified the text as requested (Line 258-260).

References:

1. Ha, G. *et al.* TITAN: inference of copy number architectures in clonal cell populations from tumor whole-genome sequence data. *Genome Res* **24**, 1881-93 (2014).
2. Murphy, A.J. *et al.* Forty-five patient-derived xenografts capture the clinical and biological heterogeneity of Wilms tumor. *Nature Communications* **10**, 5806 (2019).
3. Gadd, S. *et al.* Genetic changes associated with relapse in favorable histology Wilms tumor: A Children's Oncology Group AREN03B2 study. *Cell Rep Med* **3**, 100644 (2022).
4. Bock, C. *et al.* High-content CRISPR screening. *Nature Reviews Methods Primers* **2**, 8 (2022).
5. Love, M.I., Huber, W. & Anders, S. Moderated estimation of fold change and dispersion for RNA-seq data with DESeq2. *Genome Biology* **15**, 550 (2014).
6. Barta, J.A., Pauley, K., Kossenkov, A.V. & McMahan, S.B. The lung-enriched p53 mutants V157F and R158L/P regulate a gain of function transcriptome in lung cancer. *Carcinogenesis* **41**, 67-77 (2020).

7. Ooms, A.H. *et al.* Significance of TP53 Mutation in Wilms Tumors with Diffuse Anaplasia: A Report from the Children's Oncology Group. *Clin Cancer Res* **22**, 5582-5591 (2016).
8. Dix, D.B. *et al.* Treatment of Stage IV Favorable Histology Wilms Tumor With Lung Metastases: A Report From the Children's Oncology Group AREN0533 Study. *J Clin Oncol* **36**, 1564-1570 (2018).
9. Coutinho, D.F. *et al.* Validation of a non-oncogene encoded vulnerability to exportin 1 inhibition in pediatric renal tumors. *Med (N Y)* **3**, 774-791.e7 (2022).
10. Young, M.D. *et al.* Single cell derived mRNA signals across human kidney tumors. *Nature Communications* **12**, 3896 (2021).

Reviewers' comments:

Reviewer #1 (Remarks to the Author):

The authors have performed additional experiments and responded to all my questions. But with new data, more inconsistencies appear that are striking when comparing the old and new versions, raising doubts about the credibility of all the data. The paper still is not telling a coherent story. Although the manuscript was revised, it is not carefully prepared regarding data labeling, links to figures or suppl. material.

In detail, with regard to:

Point 1:

Culture methods are now described sufficient and clearly.

Points 2 – 4:

Variant positions depend on the transcript used for annotation. It is not sufficient to give only coding and aa position without indicating which transcript the data refers to. Give genome positions, otherwise the data cannot be uniquely assigned.

Multiple mutations given in the initial manuscript in Fig.1 are missing now. What happened to them? There are 7/20 empty lines in the table (ARID1B, ATRX, KMT2D, MYCN, TNRC18, WTX, XPO5). Could the mutations indicated previously not be validated? LOH 17p is indicated in suppl table S1, but no data is shown.

The authors should not include SNPs (e.g. DROSHA P100L, TP53 V157, NOTCH1 G661S, PAX5 S213L, ...) in Fig. 1. This suggests well known tumor driver mutations (eg DROSHA E1147K) in the corresponding tumors that are not present. The readers can only get this clearly if they check the supplement. There are many discrepancies between the alterations given in the initially submitted manuscript and the updated version:

- PEDS0002: initially 12p/q gain, ARID1V + ATRX + KMTD2 missense mutation; updated data no alteration -> Was the old data not trustworthy? There is no proof of tumor origin of this cell line, could be normal cells but the cell line is used in multiple experiments (CRISPR, RNAi screening, ...)
- PEDS0022: 4/5 of the initially given alterations appear in the updated version. Two of them (NOTCH1 Gly661Ser, PAX5 S213L) are common population SNPs (see Gnomad data base) present in control and tumor tissue, indicating patients specific SNPs, not somatic tumor specific alterations that proof tumor origin of the cell culture/PDX. MYC and PIK3CA mutations occur only in cell culture, but are absent from tumor tissue (variant assignment is not possible due to missing information on position)...
- PEDS0023: initially 1q gain in culture + PDX, 12p/q loss, DROSHA + MYCN + TP53 + XPO5 missense; updated 1p LOH; 1q gain in cell culture, 1q loss in PDX; DROSHA + TP53 missense -> multiple discrepancies! DROSHA P100L and TP53 V157F are common SNPs and both are present in normal and tumor tissue, they are not tumor drivers.
- PEDS0041: initially 12 p/q gain, CREBBP + FGFR1 + NOTCH2 alterations in tumor cell line; updated the same alterations and additional 1p LOH, that was not mentioned before. Classified as tumor derived cell line, with 1p LOH being the only discrepancy between cell line and PDX cell line.
- PEDS1012: initially BCOR stop + TNRC18 missense, update: 1q loss, BCOR stop + PIK3CA missense -> only the truncating BCOR mutation is consistent.
- Aflac2315: initially no alterations; update 1p and 16q LOH but the exact regions differ between tumor tissue and cell culture, 18q gain in tumor tissue but not tumor cell line.
- Aflac2365: initially 12p/q gain, DROSHA missense; update 1p + 16q LOH in culture but not tumor tissue, DROSHA E1147K in tumor and cell line
- Aflac2377: initially 17p gain, MED12+MYC missense, WT1 truncation; update 18q loss, WT1 truncation, MED12 + MYC missense, but MYC and 18q alteration in cell line only and absent from

tumor tissue.

- Aflac2494 and Aflac2597 are additional cell lines. 2494 harbours tumor specific WT1 and CTNNB1 alterations and they are well characterized as tumor cells. 2597 showed CNV at chr 1 and 12p, but regions differ between tumor tissue and cell line and the BCOR missense mutation is present in tumor tissue only while corresponding cell line has wildtype BCOR. Some of these differences are not visible in fig1 and they do not classify Afl2597 as tumor cell line.

There is not a single tumor/cell line/PDX where the data given initially and now in the updated version are consistent.

There are only four of ten cell lines (PEDS0041, Afl2365, Afl2377, Afl2494) with consistent typical WT driver alterations.

Point 5:

Fig 2a clearly marks tumor tissue and cell cultures and shows clustering of WT cell within the WT cohort. But one cannot decide whether they cluster with the tumor tissue they were derived from.

Point 6: I fully agree that the growth rate, limited proliferation time and thus limited cell number are a concern when using primary cell lines. PEDS0002 showed no tumor specific alterations, thus tumor origin could not be proven, and the cell culture could consist of normal fibroblast or other normal kidney cells. PEDS0023 culture and PDX harboured no consistent WT driver mutation, and they differ in 1q status. So, it is somewhat arbitrary to compare PEDS0002 and 0023 and call them representative WT cultures for CRISPR-Cas9 screening.

CRISPR screening: suppl figS1e (= previous fig2b) suggests 58 (51+6+1) PEDS0002 specific genes, not 60 as stated in figure 3b.

Point 7: okay

Point 8: "We have added the recommended Fig 2e to reflect XPO1 expression in normal tissue, in normal kidney cultures, in tumor samples and corresponding tumor cell cultures."

It's 2b, not fig. 2e. This figure clearly shows XPO1 expression at the same level in normal tissue/cell cultures and tumor tissue/cell cultures used.

Point 9: I understand that it is difficult to obtain usable normal kidney cell cultures for many reasons. It is important and good that the authors were able to test 2 additional cultures. It shows that the kidney cultures respond very differently to the treatment, and some are similarly sensitive to tumor cultures.

„Finally, we have added the dose response curves for all cultures used in Supp Figure 4". It is suppl fig S3!

Point 10: Okay, all requested data is shown.

FigS2a: There is no correlation of XPO1 expression and KPT treatment.

Fig S2c+d: Which concentration of KPT is used for tumor cell culture treatment? 1012T and 0041T are represented in both, S2c and S2d but look different. Normal kidney cells Afl2494N and Afl2597N showed the same accumulation of TP53 as did tumor PEDS1012T.

TP53 V157F is considered tolerated according to the TP53 database, it's not an oncogenic variant. Otherwise, patient PEDS0023 would suffer from Li-Fraumeni syndrome, as the variant is already present in normal tissue.

Point 11: All requested details were indicated.

Point 12: Okay

Point 13: All requests are addressed.

Point 14: Authors discussed and explained the topic.

PEDS0023 is derived from a DAWT, but TP53 V157F mutation is unlikely the pathognomonic alteration, as it's predicted to be tolerated and is already a germline variant in this patient (see above). 0023T may harbour an additional TP53 alteration not detected.

Point 15: Fig4b is supposed to represent the same data as fig3b in the initial manuscript. But numbers differ by several orders of magnitude. Max. $-\log(p\text{value})$ is 4 in the old volcano plot, but 30 in the updated version. How can such large variations occur? The methods used (DESeq2) are the same in both manuscript versions and analysis is based in the same raw data. In contrast fig3f, now fig4f, stayed the same.

There is mislabelling in fig3d: There are no patients with PEDS_2494 or PEDS_2597.

Why protein data of Afl2365 is missing from fig3e, that was present in former fig2e?

The authors stated „However, ref 42 used HEK293Ts to perform their TRIP13 overexpression studies whereas we are using Wilms tumor cell lines to assess the role of TRIP13. It is possible that TRIP13 overexpression has differing roles in different lineages and contexts.“ That's not true. HEK293 cells were used in this study [now ref 48] to produce virions for transfection only. Yost et al clearly showed that LoF mutations cause chromosome segregation errors and spindle assembly checkpoint deficiency in WT-patient derived lymphoblasts and HCT116 cells and can be rescued by overexpression of wildtype TRIP13. I agree, that TRIP13 may have different roles in different cell contexts. Therefore, it is of great importance to characterize the cell models used carefully (see points 2-4).

It's clearly proven that TRIP13 is a cancer predisposition gene and biallelic loss-of-function mutations confer a high risk of WT and predispose to chromosome segregation dysfunction. So why should reduction of TRIP13 be beneficial for cancer patients?

It remains unexplained, why TRIP13 is selected for further studies. There are about 50 G2/M checkpoint genes downregulated in KPT330 treated PEDS0041 cells...

Point 17: Why is the new fig 4g-h not equal to corresponding old fig3g-h? Gene set enrichment analysis (GSEA) enrichment score curves for the "RB1/RBL1 skin specific knockout" is missing, instead "E2F targets" is in. In Fig4h (old fig3h) gene numbers changed dramatically. Previously 8 + 4 genes were deregulated in both, KPT and shTRIP treated cells – now 66 overlapping genes are indicated. Total number of deregulated genes is increased, too.

Point 18: The authors explained clearly why doxorubicin was chosen for synergy experiments.

Information on used concentrations of vincristine and actinomycin is missing. Detailed data for these two agents are not shown as for doxorubicin in Figure 5b, but should be shown (at least in supplement).

Doxorubicin and actinomycin have the same mechanism of action; both are intercalating agents that inhibit topoisomerase as well as DNA and RNA polymerases. Do the authors have an explanation for why they found different effects in co-treatment experiments?

Point 19: "We thank the reviewer for identifying this. We agree with the reviewer this data was confusing. As such, we have removed sub-panel 4b."

One cannot exclude existing data that does not fit into the proposed model. Old Fig4b is the basis for calculation of synergy scores depicted in new fig 5a. This "raw" data cannot be excluded if used for

analysis. The corresponding data should also be depicted for vincristine and actinomycin.

"We have since included all 12 tumor cell lines and all 5 normal cell lines in ... Fig 5b, and Fig 5c."
Some of these data contradict with the data shown previously. Old fig4a compared to fig5b showed: IC50 PEDS23N 1700 vs 200 nM, PEDS23T 400 vs 100 nM, PEDS22T 10 nM vs 200 nM, ... PEDS22T was the most sensitive culture in the old version, now it's the least sensitive. The previous values are given in the text (line 246... IC50 normal cells 0.65-1.5 μ M, tumor cells 20-500 nM), but do not correspond to the new figure 5b. These inconsistencies limit the credibility of the data. In addition, there is mislabelling: There is no PEDS2494 as labelled in fig5b.

Additional:

Two different tables are labelled as suppl Table S4 – primers + screen data.

Reviewer #2 (Remarks to the Author):

While the original manuscript was quite meritorious for publication, the detailed responses to each of the reviewer criticisms has markedly improved the article impact. This reviewer finds merit for publication as is in Comm Bio, meeting its rigorous standards. Congratulations on a comprehensive and well-conducted study!

Reviewer #3 (Remarks to the Author):

The authors have adequately addressed my criticisms and submitted a markedly improved manuscript version. I have no further comments.

Reviewer #1 (Remarks to the Author):

In detail, with regard to:

Point 1:

Culture methods are now described sufficient and clearly.

Points 2 – 4:

Variant positions depend on the transcript used for annotation. It is not sufficient to give only coding and aa position without indicating which transcript the data refers to. Give genome positions, otherwise the data cannot be uniquely assigned.

Multiple mutations given in the initial manuscript in Fig.1 are missing now. What happened to them? There are 7/20 empty lines in the table (ARID1B, ATRX, KMT2D, MYCN, TNRC18, WTX, XPO5). Could the mutations indicated previously not be validated? LOH 17p is indicated in suppl table S1, but no data is shown.

The authors should not include SNPs (e.g. DROSHA P100L, TP53 V157, NOTCH1 G661S, PAX5 S213L, ...) in Fig. 1. This suggests well known tumor driver mutations (eg DROSHA E1147K) in the corresponding tumors that are not present. The readers can only get this clearly if they check the supplement. There are many discrepancies between the alterations given in the initially submitted manuscript and the updated version:

- PEDS0002: initially 12p/q gain, ARID1V + ATRX + KMTD2 missense mutation; updated data no alteration -> Was the old data not trustworthy? There is no proof of tumor origin of this cell line, could be normal cells but the cell line is used in multiple experiments (CRISPR, RNAi screening, ...)

- PEDS0022: 4/5 of the initially given alterations appear in the updated version. Two of them (NOTCH1 Gly661Ser, PAX5 S213L) are common population SNPs (see Gnomad data base) present in control and tumor tissue, indicating patients specific SNPs, not somatic tumor specific alterations that proof tumor origin of the cell culture/PDX. MYC and PIK3CA mutations occur only in cell culture, but are absent from tumor tissue (variant assignment is not possible due to missing information on position)...

- PEDS0023: initially 1q gain in culture + PDX, 12p/q loss, DROSHA + MYCN + TP53 + XPO5 missense; updated 1p LOH; 1q gain in cell culture, 1q loss in PDX; DROSHA + TP53 missense -> multiple discrepancies! DROSHA P100L and TP53 V157F are common SNPs and both are present in normal and tumor tissue, they are not tumor drivers.

- PEDS0041: initially 12 p/q gain, CREBBP + FGFR1 + NOTCH2 alterations in tumor cell line; updated the same alterations and additional 1p LOH, that was not mentioned before. Classified as tumor derived cell line, with 1p LOH being the only discrepancy between cell line and PDX cell line.

- PEDS1012: initially BCOR stop + TNRC18 missense, update: 1q loss, BCOR stop + PIK3CA missense -> only the truncating BCOR mutation is consistent.

- Aflac2315: initially no alterations; update 1p and 16q LOH but the exact regions differ between tumor tissue and cell culture, 18q gain in tumor tissue but not tumor cell line.

- Aflac2365: initially 12p/q gain, DROSHA missense; update 1p + 16q LOH in culture but not tumor tissue, DROSHA E1147K in tumor and cell line

- Aflac2377: initially 17p gain, MED12+MYC missense, WT1 truncation; update 18q loss, WT1 truncation, MED12 + MYC missense, but MYC and 18q alteration in cell line only and absent from tumor tissue.

- Aflac2494 and Aflac2597 are additional cell lines. 2494 harbours tumor specific WT1 and CTNNB1 alterations and they are well characterized as tumor cells. 2597 showed CNV at chr 1 and 12p, but regions differ between tumor tissue and cell line and the BCOR missense mutation is present in tumor tissue only while corresponding cell line has wildtype BCOR. Some of these differences are not visible in fig1 and they do not classify Afl2597 as tumor cell line.

There is not a single tumor/cell line/PDX where the data given initially and now in the updated version are consistent.

There are only four of ten cell lines (PEDS0041, Afl2365, Afl2377, Afl2494) with consistent typical WT driver alterations.

We appreciate Reviewer #1 for detailing the number of changes made to our initial submission. In our initial revision, we added new samples and associated sequencing files as well as added sequencing of prior samples where we had gaps identified from our initial submission. All sequencing files (initial submission and revision) have been uploaded into the European Genome-Phenome Archive (EGA) under accession number EGAS00001007389.

In particular, as we reviewed our ultra-low pass WGS sequencing data, we thought that the signal to noise ratio could be improved to enhance our confidence in our calls in copy number changes. Thus we re-sequenced Aflac2315, Aflac2365, Aflac2377, CCLF_PEDS0002, CCLF_PEDS1012, CLF_PEDS0022, CLF_PEDS0023, CLF_PEDS0041 at higher depth of 1x using the same genomic isolates. We then re-analyzed the data in conjunction with our new models which accounts for the copy number changes noted by Reviewer 1.

When we were reviewing our WES data, we identified several gaps from our initial submission and so we performed WES where there were gaps as well as the new models we derived during the revision. We then re-analyzed all the VCF files for potentially clinically impactful variants using OpenCRAVAT version 2.4.2. These are detailed in our Methods section and account for the changes identified by the reviewer. To keep filtering consistent we have decided to keep mutations in Fig 1 (*DROSHA* P100L, *TP53* V157, *NOTCH1* G661S, *PAX5* S213L). As OpenCRAVAT filtering has deemed these to be likely pathogenic, it would be arbitrary to remove these and not others.

With respects to the concerns that CLF_PEDS0002T or other Wilms tumor cells are truly cancer cell lines, this is the rationale for performing RNA-sequencing and looking at how these cell lines clustered with other established pediatric kidney cancers and normal kidney from TARGET and St. Jude¹⁻³. We see in **Figure 2** that all our cell lines including CLF_PEDS0002T clustered with other Wilms tumor samples and cell lines. Further from the genomic analysis of 76 Wilms tumor samples in performed by Gadd et al, 2017 (previously referenced), they had 4/76 tumor samples (5.2%) which had no mutations observed within genes implicated in Wilms tumor (sample names: PAJNYT, PAJMJK, PAEBXA, PAJMEN). It may be that 2 of our tumor samples (CLF_PEDS_0002T and Aflac_2315) fall within this 5% of tumors with no known classical mutations associated with Wilms tumor.

To more generally address the concern about heterogeneity between normal, tumor, and matched cell lines, it is known that there is genetic mosaicism in the normal kidney tissues of patients with Wilms' Tumor⁴. This may account for why we are seeing *DROSHA* and *TP53* mutations in both the normal and tumor samples of PEDS_0023 and *NOTCH1* and *PAX5* in PEDS_0022T. Alternatively, it may be that some of tumor tissue was included within the normal tissue. However given the UMAP clustering presented in **Figure 2A** where tumor and cell lines cluster together, this is unlikely. Further it is known that there is significant tumor heterogeneity in Wilms tumor patients^{5,6}. This may account for why we are seeing differences between the cell line and tumor (e.g. *PIK3CA* mutations present only in cell line for PEDS_0022T, 1p + 16q LOH only in cell lines for Aflac_2365T, and *MYC* and 18q alteration present only in the cell line for Aflac_2377T). When deriving these cell lines, there may be shifts in tumor heterogeneity as we culture these cells leading to some of these changes. Future studies are needed to better understand this.

We have added these limitations and agree that further study of this is needed:

(lines 116-121)

We identified genetic heterogeneity between the tumor and the tumor derived cell line in 5 of the 10 patient samples (Aflac_2315, Aflac_2377, Aflac_2597, PEDS_0023, and PEDS_0041), supporting the previously observed genetic heterogeneity in Wilms tumor samples^{19,29}. Lastly we identified 2 of the 10 patient-derived tumor cell lines had no mutations typically observed in Wilms tumor, an observation also seen in 5% of patient samples in prior genomic analyses of Wilms tumor²⁶.

To further confirm that our cell lines were consistent with Wilms tumor, we performed RNA-sequencing of these samples and compared it to the TARGET²⁶ and St. Jude Children's Research Hospital's WT datasets¹⁹ (Methods) using uniform manifold approximation and projection (UMAP)³⁰."

(lines 346-350)

Another limitation of this study is the genetic heterogeneity and mosaicism identified within the tumor and normal samples used in this study. We have transcriptionally confirmed our samples and respective cell lines represent Wilms tumor or normal kidney tissue; however this does not elucidate the mechanisms by which mosaicism and genetic heterogeneity may drive disease pathogenesis in Wilms

tumor. Further studies will be needed to assess these features of Wilms tumor in *in vitro* and *in vivo* settings.

Otherwise, we have since added the genomic positions to each of the variants in **Supp Table 1** and we have since removed the empty rows in **Fig 1**.

Point 5:

Fig 2a clearly marks tumor tissue and cell cultures and shows clustering of WT cell within the WT cohort. But one cannot decide whether they cluster with the tumor tissue they were derived from.

We agree that the point of **Figure 2a** was to show that despite few genomic alterations shown in **Figure 1**, our tumor cell lines and tumor tissues cluster with two independent cohorts of Wilms tumor samples (TARGET and St. Jude) rather than clustering with other pediatric kidney cancers or with pediatric normal kidneys.

Point 6: I fully agree that the growth rate, limited proliferation time and thus limited cell number are a concern when using primary cell lines. PEDS0002 showed no tumor specific alterations, thus tumor origin could not be proven, and the cell culture could consist of normal fibroblast or other normal kidney cells. PEDS0023 culture and PDX harboured no consistent WT driver mutation, and they differ in 1q status. So, it is somewhat arbitrary to compare PEDS0002 and 0023 and call them representative WT cultures for CRISPR-Cas9 screening.

We appreciate the reviewer for this concern. As mentioned in Reviewer #1 Point 5, we used complementary studies (e.g., WGS, WES and RNAseq) to determine if this could be something other than a cancer cell line. While these samples may not be “classic” Wilms tumor cases based on genetic mutations, we have transcriptionally showed they represent Wilms tumor. Furthermore from a technical standpoint, these were the two cell lines which tolerated a focused library of CRISPR-Cas9 guides. We agree that if we are able to improve upon our cell culturing techniques in the future, that we can aim to perform new screens which can better capture the heterogeneity of Wilms tumor.

To this end, we have modified in our discussion (lines 342-343):

“In addition, limited expansion of these cell lines prevents high throughput studies across other cell lines included in this study.”

CRISPR screening: suppl figS1e (= previous fig2b) suggests 58 (51+6+1) PEDS0002 specific genes, not 60 as stated in figure 3b.

We thank the reviewer for identifying this mistake. We have since changed the values presented in **Figure 3b** to accurately represent the number of genes shared in each condition.

Point 7: okay

Point 8: “We have added the recommended Fig 2e to reflect XPO1 expression in normal tissue, in normal kidney cultures, in tumor samples and corresponding tumor cell cultures.”

It's 2b, not fig. 2e. This figure clearly shows XPO1 expression at the same level in normal tissue/cell cultures and tumor tissue/cell cultures used.

We appreciate the reviewer for identifying the similar expression of *XPO1* within the normal and tumor cell lines used in this study. We have since revised the manuscript to reflect this in lines 128-131:

“Interestingly, a gene which has a known therapeutic target, *XPO1*, was upregulated across renal tumors (**Fig 2b**). Although the tumor cell lines exhibited similar expression of *XPO1* as compared to tumor tissue, we also found that the normal cell lines included in this study also had upregulation of *XPO1*.”

Point 9: I understand that it is difficult to obtain usable normal kidney cell cultures for many reasons. It is important and good that the authors were able to test 2 additional cultures. It shows that the kidney cultures

respond very differently to the treatment, and some are similarly sensitive to tumor cultures.
„Finally, we have added the dose response curves for all cultures used in Supp Figure 4“. It is suppl fig S3!

We thank the reviewer for identifying this typographical error, and we apologize for any confusion.

Point 10: Okay, all requested data is shown.

FigS2a: There is no correlation of XPO1 expression and KPT treatment.

Fig S2c+d: Which concentration of KPT is used for tumor cell culture treatment? 1012T and 0041T are represented in both, S2c and S2d but look different. Normal kidney cells Afl2494N and Afl2597N showed the same accumulation of TP53 as did tumor PEDS1012T.

TP53 V157F is considered tolerated according to the TP53 database, it's not an oncogenic variant. Otherwise, patient PEDS0023 would suffer from Li-Fraumeni syndrome, as the variant is already present in normal tissue.

We appreciate the reviewer for identifying the need for additional details regarding this experiment. The cells used in Supp Fig 4c and 4d (the reviewer mentioned FigS2c+d) were treated with 5uM of KPT-330 for 24 hours. We have since added these details to the Figure legend for **Supp Fig 4** (lines 849-851):

“(c) Cell lines were treated with 5μM of KPT-330 for 24 hours. XPO1/CRM levels are suppressed with accumulation of TP53 across a majority of cell lines with exception to PEDS_0023T which harbors a TP53 mutation. (d) Normal cell lines were treated with 5μM of KPT-330 for 24 hours and accumulation of TP53 was assessed.

To further answer the reviewer's concern, we have quantified the increase in TP53 levels in response to KPT-330 treatment and show this in **Fig S4e**. Across all immunoblots shown (excluding PEDS_0023T in Fig S4c), we see a significant accumulation of p53 when tumor cells are treated with KPT-330 as opposed to normal cells.

We have since added these details to the Figure legend for **Supp Fig 4** (lines 852-854):

(e) Immunoblots from Supp Fig 4c and 4d were quantified to determine the accumulation of p53 relative to β-actin, excluding PEDS_0023T. ** p-value<0.005, all comparisons represent a Student's t-test.”

Point 11: All requested details were indicated.

Point 12: Okay

Point 13: All requests are addressed.

Point 14: Authors discussed and explained the topic.

PEDS0023 is derived from a DAWT, but TP53 V157F mutation is unlikely the pathognomonic alteration, as it's predicted to be tolerated and is already a germline variant in this patient (see above). 0023T may harbour an additional TP53 alteration not detected.

Although the reviewer is concerned that the TP53 V157F mutation is unlikely pathognomonic, a prior study has shown that the TP53 V157F mutation introduces gene expression changes associated with cell viability, migration and invasion⁷. This suggests that this mutation at least in part contributes to the transformation of the tumor, however we agree that it is possible that additional genetic alterations are needed to push it completely towards transformation.

Point 15: Fig4b is supposed to represent the same data as fig3b in the initial manuscript. But numbers differ by several orders of magnitude. Max. $-\log(\text{pvalue})$ is 4 in the old volcano plot, but 30 in the updated version. How can such large variations occur? The methods used (DESeq2) are the same in both manuscript versions and analysis is based in the same raw data. In contrast fig3f, now fig4f, stayed the same.

We apologize for the lack of clarity in our initial response to the reviewers. We realized during our revision process that our collaborators used limma whereas we used DESeq2 to identify differentially expressed genes. In our revised manuscript, we used DESeq2 consistently to perform the analysis for **Fig 4b** and **4f** (initially **Fig 3b** and **3f**) – this resulted in the differences seen.

There is mislabelling in fig3d: There are no patients with PEDS_2494 or PEDS_2597.

We appreciate identifying this typographical mistake. It has since been corrected in the revised manuscript as Aflac_2494 and Aflac_2597.

Why protein data of Afl2365 is missing from fig3e, that was present in former fig2e?

We used all our aliquots of this tumor cell line performing other experiments necessary for revisions. As a result, we were unable to repeat this particular experiment for this cell line and thus do not show this in **Fig 3e** in our first set of revisions. We hope that completion of the protein data presented with other tumor cell lines which better emphasize the differences in TRIP13 levels upon KPT-330 treatment is adequate.

The authors stated „However, ref 42 used HEK293Ts to perform their TRIP13 overexpression studies whereas we are using Wilms tumor cell lines to assess the role of TRIP13. It is possible that TRIP13 overexpression has differing roles in different lineages and contexts.“ That’s not true. HEK293 cells were used in this study [now ref 48] to produce virions for transfection only. Yost et al clearly showed that LoF mutations cause chromosome segregation errors and spindle assembly checkpoint deficiency in WT-patient derived lymphoblasts and HCT116 cells and can be rescued by overexpression of wildtype TRIP13. I agree, that TRIP13 may have different roles in different cell contexts. Therefore, it is of great importance to characterize the cell models used carefully (see points 2-4).

It’s clearly proven that TRIP13 is a cancer predisposition gene and biallelic loss-of-function mutations confer a high risk of WT and predispose to chromosome segregation dysfunction. So why should reduction of TRIP13 be beneficial for cancer patients?

We apologize for our error in the description of the cells used in Yost et al. It is indeed interesting we see such differences between patient derived lymphoblasts and the TRIP13 deleted HCT116 cells from Yost et al. and patient derived Wilms tumor cells from this study. However, in these two scenarios we are using different cell types to investigate the function of TRIP13. A recent search of the literature has shown the role of TRIP13 may be cell context dependent, and specifically seems to be driving oncogenicity in numerous tumors such as glioblastoma, colorectal carcinoma, osteosarcoma, non-small cell lung cancer, and hepatocellular carcinoma⁸⁻¹². One of these studies showed in two colorectal cancer cell lines that TRIP13 suppression inhibits oncogenicity *in vitro* and tumorigenicity *in vivo*⁹. Notably, one of the cell lines used in this study is HCT116, the same cell line used in the study by Yost et al. These differing conclusions on the function of TRIP13 in the same cancer type (and cell line) perhaps even reinforce the rationale for the further studies of TRIP13 function in Wilms tumor. While Yost et al clearly showed lymphoblasts with biallelic LOF mutations have elevated mitotic exit, they did not use primary patient tumors and they did not directly assess proliferation. Additionally, this study isolated cells from patients with MVA. While these patients do have an elevated risk of developing Wilms’ tumor, this is not the patient population we derived cells lines for this study.

We have modified the following language (lines 216-222):

Previous observations in patient derived lymphoblasts from a patient with Wilms tumor which harbor a loss of function mutation in *TRIP13* have reported a decrease in proliferation upon *TRIP13* overexpression. Other studies looking at cancers such as glioblastoma, colorectal carcinoma, osteosarcoma, non-small cell lung cancer, and hepatocellular carcinoma have shown that increased *TRIP13* expression led to increased proliferation, migration, and invasion (citations 51-55 in the manuscript). These findings suggest *TRIP13* has differing roles in different lineages and contexts. Here, in our tumor cells which do not harbor *TRIP13* mutations, we see that suppression of *TRIP13* leads to decreased viability.

It remains unexplained, why *TRIP13* is selected for further studies. There are about 50 G2/M checkpoint genes downregulated in KPT330 treated PEDS0041 cells...

We had initially picked *TRIP13* as a candidate gene due to our initial differential expression analysis using limma. Using limma, *TRIP13* was the most significantly downregulated transcript associated with G2M arrest upon KPT-330 treatment whereas with DESeq2 analyses, *TRIP13* remained significant but not the top candidate. However, given its oncogenic roles in all 5 studies mentioned in the previous comment as well as kidney biology (as noted above and in lines 216-222), we subsequently pursued this gene for further studies.

Point 17: Why is the new fig 4g-h not equal to corresponding old fig3g-h? Gene set enrichment analysis (GSEA) enrichment score curves for the “RB1/RBL1 skin specific knockout” is missing, instead “E2F targets” is in. In Fig4h (old fig3h) gene numbers changed dramatically. Previously 8 + 4 genes were deregulated in both, KPT and shTRIP treated cells – now 66 overlapping genes are indicated. Total number of deregulated genes is increased, too.

Upon reanalysis of this data, we wanted to include the E2F targets gene set instead of the RB1/RBL1 skin specific knockout as the former was a more general gene set related to cell cycling. Moreover, we found that RB1/RBL1 knockout was epidermal-specific mouse knockout which is not as contextually relevant to our study in kidney. Both gene sets were significantly downregulated within our analysis, however we thought it would be better to include E2F targets gene set instead of the RB1/RBL1 skin specific knockout gene set as E2F is more classically used.

The number of differentially expressed genes in each condition has changed because we have changed the threshold for what we consider significant. We had previously used a more stringent logFoldChange cut-off of |2|. However to be more inclusive, we dropped this cut-off to a logFoldChange of |1|.

Point 18: The authors explained clearly why doxorubicin was chosen for synergy experiments. Information on used concentrations of vincristine and actinomycin is missing. Detailed data for these two agents are not shown as for doxorubicin in Figure 5b, but should be shown (at least in supplement). Doxorubicin and actinomycin have the same mechanism of action; both are intercalating agents that inhibit topoisomerase as well as DNA and RNA polymerases. Do the authors have an explanation for why they found different effects in co-treatment experiments?

We thank the reviewer for identifying the omission of detailed concentrations in this experiment. We have since added these details to the method section in lines (473-475):
“WT cells were treated with either doxorubicin (62.5 - 500nM), vincristine (62.5 - 500nM), or actinomycin (62.5 - 500nM) in combination with KPT-330 (1.25 - 10 μ M).”

Pertaining to the reviewer’s question about differential effects of doxorubicin and actinomycin, these two drugs do not carry similar efficacy. Clinically, actinomycin D is not as effective as doxorubicin in decreasing the risk for relapse in patients with high-risk Wilms tumor. For patients enrolled on Children’s Oncology Group Renal Tumor trials, patients with low-risk Wilms Tumor are treated with actinomycin D and vincristine. Doxorubicin is added for patients with high-risk Wilms Tumor¹³. More

broadly, the use of doxorubicin is generally reserved for higher risk patients with leukemias, sarcomas and Wilms tumor¹⁴. Further study will be needed to determine why we are only seeing synergy with doxorubicin as compared to actinomycin D. However, that is outside the scope of this study.

Point 19: “We thank the reviewer for identifying this. We agree with the reviewer this data was confusing. As such, we have removed sub-panel 4b.”

One cannot exclude existing data that does not fit into the proposed model. Old Fig4b is the basis for calculation of synergy scores depicted in new fig 5a. This “raw” data cannot be excluded if used for analysis. The corresponding data should also be depicted for vincristine and actinomycin.

The synergy calculations were performed on data produced from entirely independent experiments unrelated to the data presented in the original Fig 4b. We have clarified within our methods section how these experiments were performed [lines 473-481]. Given concerns from Reviewer #1 about the initially submitted Fig 4b, we tried to replicate this experiment and were unable to consistently do so across two independent researchers. As such, this is the rationale for removing the old Fig 4b. Further, we performed new independent experiments in 5a in our prior revision to address questions for the role of vincristine and actinomycin.

“We have since included all 12 tumor cell lines and all 5 normal cell lines in ... Fig 5b, and Fig 5c.”
Some of these data contradict with the data shown previously. Old fig4a compared to fig5b showed: IC50 PEDS23N 1700 vs 200 nM, PEDS23T 400 vs 100 nM, PEDS22T 10 nM vs 200 nM, ... PEDS22T was the most sensitive culture in the old version, now it's the least sensitive. The previous values are given in the text (line 246... IC50 normal cells 0.65-1.5 μ M, tumor cells 20-500 nM), but do not correspond to the new figure 5b. These inconsistencies limit the credibility of the data.

We thank the reviewer for noticing these differences between our previous and current submission. Based on Reviewer 1's comments from the previous submission, we repeated and verified many of the critical experiments to ensure rigor and reproducibility. The data shown are of biological replicates and bars show the standard deviation from these experiments (**Fig 5b**). Despite these changes, the significance to sensitivity to doxorubicin between tumor and normal cells remains the same.

In lines 254-256:

“We then evaluated doxorubicin sensitivity in these patient-derived WT cell lines. We found that all WT were sensitive to doxorubicin with IC50s in the nanomolar range (~42-256 nM) as compared to the normal cell lines which had an IC50 range of 131- 516 nM (**Fig 5b**; p-value 0.0134).”

In addition, there is mislabelling: There is no PEDS2494 as labelled in fig5b.

We thank the reviewer for identifying this typographical error. We have since revised this in the resubmitted manuscript to Aflac_2494.

Additional:

Two different tables are labelled as suppl Table S4 – primers + screen data.

We thank the reviewer for noticing this error. We have since corrected the supplementary table numbering.

Reviewer #2 (Remarks to the Author):

While the original manuscript was quite meritorious for publication, the detailed responses to each of the reviewer criticisms has markedly improved the article impact. This reviewer finds merit for publication as is in Comm Bio, meeting its rigorous standards. Congratulations on a comprehensive and well-conducted study!

We thank the reviewer for their supportive comments.

Reviewer #3 (Remarks to the Author):

The authors have adequately addressed my criticisms and submitted a markedly improved manuscript version. I have no further comments.

We thank the reviewer for their supportive comments.

1. Gadd, S. *et al.* A Children's Oncology Group and TARGET initiative exploring the genetic landscape of Wilms tumor. *Nat Genet* **49**, 1487-1494 (2017).
2. McLeod, C. *et al.* St. Jude Cloud: A Pediatric Cancer Genomic Data-Sharing Ecosystem. *Cancer Discov* **11**, 1082-1099 (2021).
3. Gadd, S. *et al.* Genetic changes associated with relapse in favorable histology Wilms tumor: A Children's Oncology Group AREN03B2 study. *Cell Rep Med* **3**, 100644 (2022).
4. Chao, L.Y. *et al.* Genetic mosaicism in normal tissues of Wilms' tumour patients. *Nat Genet* **3**, 127-31 (1993).
5. Cresswell, G.D. *et al.* Intra-Tumor Genetic Heterogeneity in Wilms Tumor: Clonal Evolution and Clinical Implications. *EBioMedicine* **9**, 120-129 (2016).
6. Murphy, A.J. *et al.* Forty-five patient-derived xenografts capture the clinical and biological heterogeneity of Wilms tumor. *Nature Communications* **10**, 5806 (2019).
7. Barta, J.A., Pauley, K., Kossenkov, A.V. & McMahon, S.B. The lung-enriched p53 mutants V157F and R158L/P regulate a gain of function transcriptome in lung cancer. *Carcinogenesis* **41**, 67-77 (2020).
8. Zhang, G. *et al.* TRIP13 promotes the cell proliferation, migration and invasion of glioblastoma through the FBXW7/c-MYC axis. *British Journal of Cancer* **121**, 1069-1078 (2019).
9. Sheng, N. *et al.* TRIP13 promotes tumor growth and is associated with poor prognosis in colorectal cancer. *Cell Death & Disease* **9**, 402 (2018).
10. Yu, D.-C. *et al.* TRIP13 knockdown inhibits the proliferation, migration, invasion, and promotes apoptosis by suppressing PI3K/AKT signaling pathway in U2OS cells. *Molecular Biology Reports* **49**, 3055-3064 (2022).
11. Lu, R. *et al.* Upregulation of TRIP13 promotes the malignant progression of lung cancer via the EMT pathway. *Oncol Rep* **46**(2021).
12. Yao, J. *et al.* Silencing TRIP13 inhibits cell growth and metastasis of hepatocellular carcinoma by activating of TGF- β 1/smad3. *Cancer Cell International* **18**, 208 (2018).
13. Green, D.M. *et al.* Treatment of Wilms tumor relapsing after initial treatment with vincristine and actinomycin D: a report from the National Wilms Tumor Study Group. *Pediatr Blood Cancer* **48**, 493-9 (2007).
14. Pritchard-Jones, K. *et al.* Omission of doxorubicin from the treatment of stage II-III, intermediate-risk Wilms' tumour (SIOP WT 2001): an open-label, non-inferiority, randomised controlled trial. *Lancet* **386**, 1156-64 (2015).

Reviewers' comments:

Reviewer #1 (Remarks to the Author):

The authors have discussed the critical points raised. However, the main problem of inconsistent data and the exclusion of some data remains.

Points 2-4:

OpenCravat is a prediction tool, that takes multiple data sources into account. But the tool cannot decide if a mutation has functional impact. If a mutation is not tumor specific (present already in normal control DNA) and appears with high frequency in the normal healthy population, it is probably a patient specific SNP and not a tumor driver mutation. According to Gnomad these variants are common: DROSHA P100L AF 2.87e-4, TP53 V157I AF 3.98e-5, NOTCH1 G661S AF 3.61e-4, PAX S213L AF 6.08e-4. Especially the TP53 alteration is well studied and classified as benign. None of these mutations are known to cause tumor predisposition. One should not include these variants in the figure, as this would lead others to consider these mutations as new WT driver events. E.g. figure1 suggests that Aflac2365 and PEDS0023 both were driven by tumor specific DROSHA mutations – but they were not. While Aflac2365 had an established WT DROSHA variant, patient PEDS0023 just harbours a DROSHA SNP.

18p is still represented in the figure but no event is reported for this region.

Point 5: I would like to know where the normal kidney cell cultures used are grouped in the transcriptome analysis (UMAP blot, Fig. 2a). Fig 2b shows that there is RNAseq data of 4 normal kidney cell cultures. Why are they not included in Fig2a?

Point 8:

Why would normal cells and tumor cells behave differently when treated with an XPO1 inhibitor if they express XPO1 at the same level? Under these conditions, XPO1 would not be considered a good tumor-specific target.

Point 10:

The increase in TP53 after KPT treatment of normal kidney cells is not significant (Fig. S4e)? What is the p-value for normal kidney cells? Readers of the manuscript will have to trust the calculation, but the WBs in Fig. S4d and the ratio blot in Fig. S4e look like there is an increase in TP53 levels, even if it is lower than in tumor cells.

Point 15: “Why protein data of Afl2365 is missing from fig3e, that was present in former fig2e? ... “

So why did the authors redo the western blot from the first submission to the 1st revision? The same cell cultures are shown, the information/data in the figures are the same, the blots look pretty much the same. But Afl2365 is not one of the “tumor cell lines which better emphasize the differences in TRIP13 levels upon KPT-330 treatment“? The data that was collected should not be omitted, but included, even if it does not support the hypothesis.

Point 17:

The data in Supplementary Table 7 appears to be manually curated - the data for the two enriched gene sets shown (HALLMARK_G2M_CHECKPOINT and HALLMARK_E2F_TARGETS) are in a surprising position in the table, pval and padj differ by several orders of magnitude from all other gene sets, but are both exactly the same ($1E-50$, $2.5E-49$, surprisingly without decimal places), no log2error is given...

Why do the p-values differ by tens of orders of magnitude between GSEA for KPT treated (suppl table 7, fig 4c) and shTRIP13 treated cells (Suppl table 9, fig 4g) with min p-value $1E-50$ vs. 0.0017 ? Is this again caused by different analyses with limma and DESeq2? The authors should urgently be consistent in their analyses. By the way: RB1/RBL1 skin specific knockout gene set is not present in the GSEA shown (suppl table 9). It remains unclear where this gene set enrichment came from in the initial submission and why it was presented anyway.

Point 18: "Detailed data for these two agents are not shown as for doxorubicin in Figure 5b, but should be shown (at least in supplement)."

No information is given on the requested data. No "raw data" on cell viability with different treatments are given (but the data should have been collected as they form the basis for the synergy calculation), and it remains completely puzzling why there is a strong synergistic effect of doxorubicin plus KPT but no synergy for actinomycin or vincristine plus KPT. Only synergy scores based on a model are presented.

Point 19: I assume that the authors already used valid data and biological replicates for the initial submission. Do the authors now no longer trust the data that they originally found good enough for publication? It is always good to repeat experiments. But these should all be included and presented as replicates.

Reviewer #1 (Remarks to the Author):

Points 2-4:
OpenCravat is a prediction tool, that takes multiple data sources into account. But the tool cannot decide if a mutation has functional impact. If a mutation is not tumor specific (present already in normal control DNA) and appears with high frequency in the normal healthy population, it is probably a patient specific SNP and not a tumor driver mutation. According to Gnomad these variants are common: DROSHA P100L AF 2.87e-4, TP53 V157I AF 3.98e-5, NOTCH1 G661S AF 3.61e-4, PAX S213L AF 6.08e-4. Especially the TP53 alteration is well studied and classified as benign. None of these mutations are known to cause tumor predisposition. One should not include these variants in the figure, as this would lead others to consider these mutations as new WT driver events. E.g. figure1 suggests that Aflac2365 and PEDS0023 both were driven by tumor specific DROSHA mutations – but they were not. While Aflac2365 had an established WT DROSHA variant, patient PEDS0023 just harbours a DROSHA SNP.
18p is still represented in the figure but no event is reported for this region.

We thank reviewer 1 for these points and have removed the following from **Fig 1**:
-DROSHA P100L
-TP53 V157I
-NOTCH1 G661S
-PAX S213L

We have removed CNA changes to chromosome 18p per Reviewer 1's request.

Please find below the revised **Fig 1**:

We additionally have removed lines 101-103 as this described the *TP53* mutation that Reviewer 1 is concerned about.

Point 5: I would like to know where the normal kidney cell cultures used are grouped in the transcriptome analysis (UMAP blot, Fig. 2a). Fig 2b shows that there is RNAseq data of 4 normal kidney cell cultures. Why are they not included in Fig2a?

We have added the normal kidney cell lines into our transcriptome analysis (revised **Fig 2a** below):

To reflect this additional data, we have added the following text to the manuscript to reflect the addition of the normal cell lines (lines 112-116):

“However, one FHWT cell line (Aflac2315) did not clearly cluster with our tumor tissue and cell lines. *SIX2* is elevated in Wilms Tumor and the mean log₂ counts for our normal tissue and cell lines were 2.54 (standard deviation of 0.32) whereas in our FHWT was 3.73 (standard deviation of 0.20). Aflac2315 had *SIX2* log₂ counts of 3.86 in the tumor and 3.39 in the cell line. More broadly, we observed that *SIX2* and *CITED1* were generally upregulated across our WT cell lines, further consistent with WT biology (**Fig 2b**)¹⁻³.”

Furthermore, this analysis was performed using the 1,000 most variable genes in Wilms tumor as identified from Trink et al.⁴ The following line of text was added to the Methods to clarify this (Lines 382-383):

“Specifically, we used the 1,000 most highly variable genes in Wilms tumor as previously identified.”

Point 8:

Why would normal cells and tumor cells behave differently when treated with an XPO1 inhibitor if they express XPO1 at the same level? Under these conditions, XPO1 would not be considered a good tumor-specific target.

As compared to normal tissue, tumor tissue has modestly higher *XPO1* levels (log₂fold: 0.086, pval: 0.008) **Fig s2d**. However, as the reviewer noted *XPO1* levels in culture are similar in normal and tumor cells. This may be due in part to the culturing conditions. Nonetheless, we would expect that normal cells and tumor cells would behave differently if XPO1 were a tumor-specific target. Specifically, we observed that the XPO1 inhibition efficacy significantly differs in normal vs. tumor cells **Fig 3d**. Even when comparing only tumor cells, it seems that XPO1 levels do not directly determine response to KPT-330 suggesting downstream pathways are driving

sensitivity **Supp Fig 4a**. When we look at **Fig S4b**, we see that the auto feedback loop response to the XPO1 inhibitor is significantly less in normal cells than that to the tumor cells. Moreover, our study, based on the prior findings and literature, indicates a distinct mechanism of action of XPO1 inhibitor in normal vs. tumor cells. We hypothesized that this distinction may stem from the tumor cells ability to develop a survival mechanism via upregulation of multiple pro-survival genes that lead to upregulation of pathways responsible for cancer progression. In our study we suspect that the XPO1 inhibitor is working by inhibition of one such gene, TRIP13, that is specifically upregulated in WT tumor patient samples (**Supp Fig 5d**). TRIP13 promotes cancer progression via multiple pathways such as AKT/mTOR⁵, deregulating cell cycle causing the upregulation of DNA damage repair leading to generation of chromosomal instability (CIN) and aneuploidy⁶ and activation of Notch signaling⁷. One of the mechanisms by which TRIP13 leads to CIN is by interacting with TTC5 a cofactor of P53⁸. Herein this study we see a significant increase in the upregulation of p53 of our tumor cells as compared to normal cells (**Fig S4e**) after treatment with XPO1 inhibitors. Thus, we suspect that XPO1 inhibitors are working via the TRIP13/P53 axis. Collectively, based on our findings, this would suggest that the response is different due to the requirement of XPO1 in Wilms Tumor cells as we had identified in our CRISPR-Cas9 and RNAi screens.

We have added the following text (lines 116-119):

“Interestingly, a gene which has a known therapeutic target, *XPO1*, was modestly upregulated across renal tumors (**Fig 2b, Fig S2d**). Although the tumor cell lines exhibited similar expression of *XPO1* as compared to tumor tissue, we also found that the normal cell lines included in this study also had upregulation of *XPO1*.”

We have added the following lines of text (line 817) of **Fig s2d**:

“(d) *XPO1* levels across the patient derived tissue used in this study.”

Point 10:

The increase in TP53 after KPT treatment of normal kidney cells is not significant (Fig. S4e)? What is the p-value for normal kidney cells? Readers of the manuscript will have to trust the calculation, but the WBs in Fig. S4d and the ratio blot in Fig. S4e look like there is an increase in TP53 levels, even if it is lower than in tumor cells.

We have since revised our **Fig s4e** (below) to show that although there is a slight increase in TP53 levels of normal kidney cells following KPT-330 treatment. However, this is not significant with a p-value of 0.0634.

Point 15: “Why protein data of Afl2365 is missing from fig3e, that was present in former fig2e? ... “

So why did the authors redo the western blot from the first submission to the 1st revision? The same cell cultures are shown, the information/data in the figures are the same, the blots look pretty much the same. But Afl2365 is not one of the “tumor cell lines which better emphasize the differences in TRIP13 levels upon KPT-330 treatment“? The data that was collected should not be omitted, but included, even if it does not support the hypothesis.

As pointed out by reviewer #1 in our initial submission, there were concerns that our initial immunoblot needed to be better optimized. We thawed our cells and reperformed the experiments. Because of the short term and limited availability of Aflac2365, we could not repeat this particular sample in our first revision. We have since included both sets of blots (one in **Fig 3e** and one in **Supp Fig 2c**) per the reviewer request.

The following text has been added (lines 815-816):

“(c) Additional replicate of the KPT-330 treated (5μM treat for 48 hours) Wilms tumor cells across 5 different cell lines.”

Point 17:

The data in Supplementary Table 7 appears to be manually curated - the data for the two enriched gene sets shown (HALLMARK_G2M_CHECKPOINT and HALLMARK_E2F_TARGETS) are in a surprising position in the table, pval and padj differ by several orders of magnitude from all other gene sets, but are both exactly the same (1E-50, 2.5E-49, surprisingly without decimal places), no log2error is given...

Why do the p-values differ by tens of orders of magnitude between GSEA for KPT treated (suppl table 7, fig 4c) and shTRIP13 treated cells (Suppl table 9, fig 4g) with min p-value 1E-50 vs. 0.0017? Is this again caused by different analyses with limma and DESeq2? The authors should urgently be consistent in their analyses. By the way: RB1/RBL1 skin specific knockout gene set is not present in the GSEA shown (suppl table 9). It remains unclear where this gene set enrichment came from in the initial submission and why it was presented anyway.

We thank reviewer #1 for pointing out these concerns.

- 1) The data in Supplementary Table 7 was not manually curated. We have since reached out to the author of fgsea and he pointed out that there is an estimation limit when running default conditions particularly when pval and padj values are low (e.g., significant). We have since removed this estimation limit and have included the estimated values in our revised **Supp tables 7 & 9**.
- 2) We noted in our review of these datasets, that the gene set enrichment analyses of our sh*TRIP13* samples included a deprecated option (e.g., testing a specific number of permutations). We have corrected this and note that there is no significant change in NES values but does partially address the concerns about the range of p-values identified. We have provided revised **Supp table 9**. We have also updated **Fig 4g** to reflect the changes in NES and FDR values and show them below as well. All conclusions remain unchanged.

HALLMARK E2F TARGETS	Shown in prior revision	Current revision
NES	-2.47	-2.47
FDR	0.006	4.7e-15

HALLMARK G2M CHECKPOINT	Shown in prior revision	Current revision
NES	-1.75	-1.76
FDR	0.006	7.3e-5

- 3) As noted by Reviewer #1, p-values differ for the GSEA for KPT-330 treated cells as compared to sh*TRIP13* treated cells. This is likely due to the fact that KPT-330 is targeting XPO1 which has a broad set of functions as compared to TRIP13. Functions of XPO1 include transporting critical proteins such as p53, RB1, p21, cyclin B1 and D1 out of the nucleus⁹. Furthermore, XPO1 has been found to be required for microtubule nucleation at kinetochores¹⁰. *TRIP13* has been described to have a more targeted set of functions in the release of the mitotic checkpoint complex which promotes mitotic progression¹¹. *TRIP13* has further been implicated in promoting homology directed repair through its regulation of REV7¹². Given this more targeted role of *TRIP13* in cellular division, we would expect *TRIP13* suppression to affect a smaller subset of genes than *XPO1* suppression.
- 4) We note that all analyses since our initial revision were done with DESeq2.
- 5) With regards to the RB1/RBL1 skin specific knockout gene set (e.g., RB_P107_DN.V1_UP) not being in the dataset shown in our revision version of **Fig 4g** and **Supp Table 9**, we would note that in our initial submission, we explored several different Human MSigDB Collections. In our first submission, we had used both the hallmark and oncogenic signature gene sets for the sh*TRIP13* analysis. Then during our first revision, we focused only on the hallmark gene sets which does not include RB_P107_DN.V1_UP. As such, to be consistent in our analyses in what was presented, only the hallmark gene sets were presented in both our figures and **Supp Table 9**.

Point 18: "Detailed data for these two agents are not shown as for doxorubicin in Figure 5b, but should be shown (at least in supplement)."

No information is given on the requested data. No "raw data" on cell viability with different treatments are given (but the data should have been collected as they form the basis for the synergy calculation), and it remains completely puzzling why there is a strong synergistic effect of doxorubicin plus KPT but no synergy for actinomycin or vincristine plus KPT. Only synergy scores based on a model are presented.

We thank the reviewer for pointing out this concern. We apologize for misunderstanding this point in the last response to the reviewers. Our understanding is that the reviewer is asking for our data from the current **Fig 5a** and **5c**. We have provided **Supp Table 12** and **13** which outlines the underlying data used

to calculate synergy for **Fig 5a** and **5c**, respectively. These matrices can be inputted directly into: <https://synergyfinder.fimm.fi/synergy/20240129233348623328/>

Furthermore, to be consistent with providing the details of the IC50s in **Fig 3d** (which is shown in **Supp Fig 3**), we have done the same for **Fig 5b** which is now shown in **Supp Fig 6**.

We appreciate the point that there is a lack of synergy of the XPO1 inhibitor, KPT-330, with vincristine or dactinomycin. Future studies will look to better understand this difference.

Point 19: I assume that the authors already used valid data and biological replicates for the initial submission. Do the authors now no longer trust the data that they originally found good enough for publication? It is always good to repeat experiments. But these should all be included and presented as replicates.

On our initial submission, Reviewer 1 mentioned concerns about the experimental conditions used for **initial submission Fig 4b** and also requested additional cell lines to be incorporated broadly throughout the manuscript. In our first revision, while we were successful in developing new cell lines, performing many of our experiments again and reproducing the results of our prior subpanels, we mentioned that we too had concerns with the experimental conditions for this subpanel. Since the findings of that subpanel were not central to the results of our work, we have opted to exclude this panel from our initial submission.

1. Murphy, A.J. *et al.* SIX2 and CITED1, markers of nephronic progenitor self-renewal, remain active in primitive elements of Wilms' tumor. *J Pediatr Surg* **47**, 1239-49 (2012).
2. Walz, A.L. *et al.* Recurrent DGCR8, DROSHA, and SIX homeodomain mutations in favorable histology Wilms tumors. *Cancer Cell* **27**, 286-97 (2015).
3. Wegert, J. *et al.* Mutations in the SIX1/2 pathway and the DROSHA/DGCR8 miRNA microprocessor complex underlie high-risk blastemal type Wilms tumors. *Cancer Cell* **27**, 298-311 (2015).
4. Trink, A. *et al.* Geometry of Gene Expression Space of Wilms' Tumors From Human Patients. *Neoplasia* **20**, 871-881 (2018).
5. Zhu, M.X. *et al.* Elevated TRIP13 drives the AKT/mTOR pathway to induce the progression of hepatocellular carcinoma via interacting with ACTN4. *J Exp Clin Cancer Res* **38**, 409 (2019).
6. Wang, K. *et al.* Thyroid Hormone Receptor Interacting Protein 13 (TRIP13) AAA-ATPase Is a Novel Mitotic Checkpoint-silencing Protein*. *Journal of Biological Chemistry* **289**, 23928-23937 (2014).
7. Zhou, X.Y. & Shu, X.M. TRIP13 promotes proliferation and invasion of epithelial ovarian cancer cells through Notch signaling pathway. *Eur Rev Med Pharmacol Sci* **23**, 522-529 (2019).
8. Yu, L. *et al.* TRIP13 interference inhibits the proliferation and metastasis of thyroid cancer cells through regulating TTC5/p53 pathway and epithelial-mesenchymal transition related genes expression. *Biomedicine & Pharmacotherapy* **120**, 109508 (2019).
9. Azmi, A.S., Uddin, M.H. & Mohammad, R.M. The nuclear export protein XPO1 — from biology to targeted therapy. *Nature Reviews Clinical Oncology* **18**, 152-169 (2021).
10. Torosantucci, L., Luca, M.D., Guarguaglini, G., Lavia, P. & Degrassi, F. Localized RanGTP Accumulation Promotes Microtubule Nucleation at Kinetochores in Somatic Mammalian Cells. *Molecular Biology of the Cell* **19**, 1873-1882 (2008).
11. Eytan, E. *et al.* Disassembly of mitotic checkpoint complexes by the joint action of the AAA-ATPase TRIP13 and p31(comet). *Proc Natl Acad Sci U S A* **111**, 12019-24 (2014).
12. Clairmont, C.S. *et al.* TRIP13 regulates DNA repair pathway choice through REV7 conformational change. *Nature Cell Biology* **22**, 87-96 (2020).

REVIEWERS' COMMENTS:

Reviewer #4 (Remarks to the Author):

Considering that I don't know the full story of this manuscript, I do believe Authors replied in a good way to all the points raised by reviewer 1.